# Polyethylene Terephthalate Microplastics Generated from Disposable Water Bottles Induce Interferon Signaling Pathways in Mouse Lung Epithelial Cells

**DOI:** 10.3390/nano14151287

**Published:** 2024-07-31

**Authors:** Luna Rahman, Andrew Williams, Dongmei Wu, Sabina Halappanavar

**Affiliations:** 1Environmental Health Science and Research Bureau, Health Canada, Ottawa, ON K1A 0K9, Canada; luna.rahman@hc-sc.gc.ca (L.R.); andrew.williams@hc-sc.gc.ca (A.W.); dongmei.wu@hc-sc.gc.ca (D.W.); 2Department of Biology, University of Ottawa, Ottawa, ON K1N 6N5, Canada

**Keywords:** microplastics, inhalation toxicology, environmental exposure, polyethylene terephthalate, cytotoxicity, genotoxicity, micronuclei, transcriptomic

## Abstract

Microplastics (MPs) are present in ambient air in a respirable size fraction; however, their potential impact on human health via inhalation routes is not well documented. In the present study, methods for a lab-scale generation of MPs from regularly used and littered plastic articles were optimized. The toxicity of 11 different types of MPs, both commercially purchased and in-lab prepared MPs, was investigated in lung epithelial cells using cell viability, immune and inflammatory response, and genotoxicity endpoints. The underlying mechanisms were identified by microarray analysis. Although laborious, the laboratory-scale methods generated a sufficient quantity of well characterized MPs for toxicity testing. Of the 11 MPs tested, the small sized polyethylene terephthalate (PETE) MPs prepared from disposable water bottles induced the maximum toxicity. Specifically, the smaller size PETE MPs induced a robust activation of the interferon signaling pathway, implying that PETE MPs are perceived by cells by similar mechanisms as those employed to recognize pathogens. The PETE MPs of heterogenous size and shapes induced cell injury, triggering cell death, inflammatory cascade, and DNA damage, hallmark in vitro events indicative of potential in vivo tissue injury. The study establishes toxicity of specific types of plastic materials in micron and nano size.

## 1. Introduction

Microplastics (MPs) and nanoplastics (NPs), collectively referred to as MPs here after, denote plastic fragments smaller than 5 mm [1] or finer than 1 µm [2], and are prevalent in various environmental settings. These minuscule plastic fragments primarily enter the environment during plastic manufacturing from monomers for commercial applications or through personal care products containing MPs [3]. Secondary MP pollution arises from the breakdown and degradation of larger plastic litter present in the environment due to factors such as heat, humidity, wind, and UV irradiation [4]. Research on marine MP pollution and its impact on marine life, as well as its implications on human health, has been extensively investigated. However, studying airborne MP pollution and its potential inhalation toxicity in humans poses significant challenges [5,6,7]. Airborne MPs are detected in outdoor and indoor air in densely populated cities such as those in China [8,9,10,11,12,13], the metropolitan area of Paris in France [10,14,15], Asaluyeh county in Iran [16], and the metropolitan area of Hamburg in Germany [17]. In a previous study conducted by the authors of this manuscript, the presence of several MP types in the respirable size (≤2.5 µm) range in Montreal, a metropolitan city in Canada, was reported [18]. Airborne MPs are much smaller in size than the traditional definition of 5 µm, and lighter in density, and thus can travel long distances from the point of release and remain suspended for a long period of time [19,20]. Studies have estimated that humans can inhale approximately 37,000 to 60,500 MPs per year, which is expected to increase over the years [21]. Zhang et. al., showed that collective human MP load through the respiratory route can reach up to 3.0 × 10^7^ particles per person per year [22]. The commonly detected MPs in air include nylon, pristine polyethylene (PE), chlorinated polyethylene (CPE), polypropylene (PP), polystyrene (PS), polyethylene terephthalate (PETE), poly vinyl chloride (PVC), and polyurethane (PUR) [9,12,13,17,18,19,23]. The abundance of MPs and the specific types, according to these reports, vary depending on the population density, lifestyle, sampling sites, and sampling methods. From the studies above, the reported concentrations of daily atmospheric fallout ranged from 2.1 to 602 particles·m^−2^, where the suspended MP concentrations ranged from approximately 0.3 to 174.97 particles·m^−3^. Comparatively, MP concentrations in indoor spaces are high. For example, Dris et al. found approximately 1.1 to 60 MP particles·m^−3^ in indoor air compared with approximately 0.3 to 1.5 particles·m^−3^ in Paris, France [14]. Similar results were found in cities in China where 1583 ± 1180 particles·m^−3^ and 185 ± 85 particles·m^−3^ were found in indoor and outdoor air, respectively [11]. The sources of indoor MPs included synthetic textiles, synthetic floorings, carpeting, household furniture, plastic food storage containers, plastic toys, etc. [24].

The lung serves as the primary target organ for inhalable MPs. Humans are exposed to inhalable-size MPs not only by inhaling aerosolized MPs from the environment but also through many polymer-based pulmonary drug delivery systems [25]. In recent human clinical studies, MPs have been detected in human sputum [26], bronchial lung fluid (BALF) [27], and lung tissue [28]. In these studies, MPs found in BALF and tissue were mostly fibres—97% and 49%, respectively [27,28]. An abundance of PP, PE, PETE, and resin were detected in various sections of lung tissue from eleven out of thirteen patients investigated by Jenner et al. [28]. Plastic polymers are durable and biopersistent; a study involving in vitro testing of PE, PP, and polycarbonate fibers in synthetic extracellular lung fluid showed no discernible change in the chemical or physical characteristics of the polymers post-6 months suspension in the fluid. This suggests that, once inhaled, MPs may persist in BALF for extended periods, which raises concerns for potential human health implications [29,30]. MPs from the target lung tissue may translocate to other organs. In a recent clinical study, a higher rate of infarction, nonfatal stroke, or death was observed among asymptomatic patients with internal carotid artery stenosis detected with MPs in their carotid artery plaques compared with the patients not detected with MPs in their carotid artery plaques [31]. In the past, the outbreak of interstitial lung disease among textile workers at nylon flock industries situated in Ontario, Canada and in Rhode Island, United States of America during 1992–1996 was associated with exposure to respirable-size fragmented nylon [32,33]. These results suggest a potential for tissue exposure to MPs and adverse effects in humans.

Inflammatory responses, immunosuppression, and oxidative stress have been observed in marine animals following MP exposure. PS MPs smaller than 100 µm have been found accumulated in the larvae and guts of fish and have been shown to induce oxidative stress and inflammation, thereby negatively impacting liver metabolism, the reproductive system, and neuronal development in fish [34,35,36]. The transcriptional modulation of the immune response was observed in Zebra fish muscles [37] following PE and PS MPs.

In rodents, pulmonary injury has been observed after the intratracheal instillation of 100 nm or 1 µm PS in Sprague Dawley rats and post-instillation of 5 µm PS in C57BL/6 [38] or in Institute of Cancer Research (ICR) mice [39]. Pulmonary injury has also been observed after inhalation exposure to 40 nm PS in C57BL/6 mice [40]. In addition, increases in the number of inflammatory cells and levels of inflammatory proteins were observed in BALF of C57BL/6 and ICR mice, but not in BALB/c mice [41]. Concentration-dependent increases in the levels of inflammatory proteins, transforming growth factor beta (TGF-β), and tumor necrosis factor alpha (TNF-α) were found in lung tissue in Sprague Dawley rats 14 day post-inhalation of 0.01 µm PS MPs [42]. In the same study, changes in biochemical, hematological, and respiratory function markers were also observed but in a concentration-independent manner. The differential expression of inflammatory and immune response-related genes, e.g., increased interleukin (IL) 1β, interferon (IFN), and TNF-α in C57BL/6 mice or mucin and Kruppel-like genes in ICR mice, has been observed following oral exposure to 5 μm PS MPs [39,43]. Inflammation and perturbations in metabolic homeostasis have been observed in high-fat diet obese mice [44] or in ICR mice [45] following six-weeks oral exposure to 1–5 μm PS MPs, while cardiac fibrosis was reported in male Wistar rats following 90-days oral exposure to spherical 0.5 μm PS MPs [46].

Similarly, in mammalian cell cultures, a loss of viability and an increased expression of pro-inflammatory markers and genotoxic responses have been reported post-exposure to PS MPs in human lung epithelial A549 cells [47], epithelial cells isolated from normal human bronchial epithelium derived from autopsies of noncancerous individuals (BEAS2B cells) [48], colorectal cancer cells HCT116 [49], human foreskin fibroblast (Hs27) cells [50], human lymphoblast cells (TK6) [49], human cerebral cells (T98G), and epithelial cells (HeLa) [51]. PP and PE MPs were shown to induce increased Reactive Oxygen Species (ROS) production in human dermal fibroblast cells [52] and in T98G cells [51]. The increased secretion of pro-inflammatory cytokines such as, IL-1β, IL-6, and TNF-a in macrophages (THP-1 cell lines) or IL-6, IL-8, and TNF-a in human colorectal adenocarcinoma cells (Caco-2 cells) was observed following exposure to 59 nm or 100 µm PS [53,54]. Increased cytotoxic and genotoxic effects were also observed in human lung epithelial cells (Calu-3) and THP-1 cells following exposure to 50 nm to 5 µm PS [38,55,56]. Thus, there is substantial evidence to suggest that MPs are present in the environment in an inhalable fraction and that some of them possess the potential to be harmful if exposed to a sufficient quantity via different routes of exposure.

Currently, many studies examining the toxicity of MPs primarily utilize PS particles of uniform size and mostly spherical shape, which may not accurately represent environmental exposure scenarios. Moreover, there is a scarcity of research on the toxicity of other common plastics such as PP, PE, and polyamides (PA) [57]. Additionally, the forms and shapes of these plastics used in studies often do not reflect their real-world environmental counterparts. This highlights the need for more comprehensive and representative research to better understand the potential toxicity of MPs in various environmental contexts. The primary objective of the present study was to develop laboratory-scale methods for producing a sufficient quantity of well-characterized MPs sourced from the types of plastics commonly found in the environment, for use in toxicological studies. Specifically, MPs with diameters less than 2.5 μm or smaller than 1 μm were prepared, as they have the potential to penetrate deeper into the lungs and traverse cell and tissue barriers. The selection of plastic types for generating MPs was guided by the findings of an earlier study conducted by the authors of the present study [18], which identified the types of MPs present in outdoor air.

The exact adverse outcomes induced by MPs in humans have not been fully elucidated. However, insights can be gleaned from studies examining similar substances such as engineered nanomaterials and air particulates [58]. It can be inferred that the toxicity of MPs may stem from various factors including their physical and structural properties (such as size and shape), as well as their chemical composition. This includes both the chemical composition of the plastics themselves and any chemicals that are adsorbed onto the surface of the plastic particles. This study also suggested that inhaled MPs may trigger acute cellular events in the lungs, including inflammation, oxidative stress, and cytotoxicity. As a result, a tiered testing strategy has been proposed focusing on these three acute events, as they are known to precede tissue-level injury [58]. Therefore, the second objective of the present study was to evaluate the potential of MPs to induce cytotoxicity, genotoxicity, and inflammation in lung cells under submerged conditions. In brief, immortalized lung epithelial (FE1) cells derived from the transgenic Muta Mouse rodent model were exposed to a suite of MPs of different sizes, chemical identity, and shapes. FE1 cells were specifically selected, as they allow investigation of genotoxicity including mutation frequency analysis, an important endpoint for consideration in the assessment of carcinogenic potential of MPs. Cytotoxicity, pro-inflammatory response, and micronuclei (MN) induction endpoints were assessed at different exposure concentrations of MPs. For MPs showing toxicity in more than one endpoint, whole genome microarray analysis was conducted to identify the underlying mechanisms of toxicity.

## 2. Materials and Methods

### 2.1. Commercially Purchased MPs

Aqueous suspension of unmodified PS beads of 60, 100, and 300 nm (National Institute of Standards and Technology (NIST), Gaithersburg, MD, USA); low-density polyethylene (LDPE) large beads (1000–5000 nm, non-nano certified reference material, Nanochemazone™, Alberta, Canada); high-density polyethylene (HDPE) large beads (1000–5000 nm, non-nano certified reference material, Nanochemazone™, Alberta, Canada); bulk polyamide nylon powder (5–50 µm, non-nano certified reference material, Sigma Aldrich, Mississauga, ON, Canada); and polymethyl methacrylate (PMMA, 1500–11,500 nm, Cospheric, CA, USA) were purchased. The 60, 100, and 300 nm PS suspensions contained an emulsifier (possibly, sodium lauryl sulfate or sodium 1-dodecanesulfonate and 1-dodecanol at a concentration of <0.05%) as well as an electrolyte (~0.02%) to prevent agglomeration. The 60 nm PS contained a carboxyl group, which helps with MP stabilization in the suspension. The 100 and 300 nm PS suspensions contained 50 ppm sodium azide to retard the growth of algae and bacteria. All commercially purchased materials were used as received without further purification.

### 2.2. Lab-Generation of MPs

Figure 1 is a schematic of the methods used for generating MPs from different plastic types.

#### 2.2.1. Preparation of Nylon MPs from Bulk Nylon Polyamide Powder

About 200 mg of the bulk nylon polyamide powder was sonicated in 1 L methanol (Omnisolve, 99.9%, Sigma Aldrich, Oakville, ON, Canada) in a clean autoclaved glass bottle for 20 min using a Bransonic digital ultrasonic water bath (model 2800, Branson, Danbury, CT, USA) at 40 kHz at room temperature (RT). Post-sonication, a sequential membrane filtration was conducted using a clean autoclaved membrane filtration apparatus, with the glass fibre membranes of 7 and 2.7 µm pore sizes. The solvent (methanol) was removed by application of nitrogen gas flow.

#### 2.2.2. Preparation of Nylon MPs from the Tea Bags

Fifty nylon tea bags were cut into small pieces of ~1 cm × 1 cm in size and microwaved in MilliQ water for a total of 6 h, with 10 min of microwaving followed by 20 min pause. This resulted in nylon MPs of <2.7 µm. The resulting suspension in water was frozen at −80 °C overnight and then the water was removed by freeze drying method using a lyophilizer (Labconco, Freezone Benchtop 2.5L, Kansas City, MO, USA).

#### 2.2.3. Preparation of PETE and PP MPs

Five disposable plastic water bottles made from PETE and four individual polypropylene food storage containers (PP) were washed thoroughly with soap, followed by 70% ethanol and MilliQ water. Clean plastic items were cut into 1 cm^2^ pieces, lightly sprayed with Milli Q water, and stored separately in 50 mL PP falcon tubes at −80 °C, for 3 days. Post 3-day freezing, before blending, the plastic squares were immersed in methanol (~1:2 volume ratio) in a glass beaker and kept on dry ice. Using a kitchen steel hand blender, the PETE and PP plastic pieces in methanol were blended separately until a layer of visible plastic suspension was obtained and further sonicated in an ice bath at 40 kHz for 2 h, using a Bransonic digital ultrasonic water bath sonicator (model 2800, Branson, Danbury, CT, USA) to break up the aggregates until a uniform suspension of MPs was evident. The solvent methanol was evaporated using nitrogen gas flow. This method resulted in 20 mg of PP of 1–5 µm in size. From disposable water bottles, 200 mg of PETE of ~1 µm in size (small PETE) and 20 mg of PETE of 1–5 µm in size (large PETE) were obtained. Dried MPs were stored at RT in sterile glass vials. It is important to note that the final MP yield does not reflect the volume of plastics from five bottles or food containers. The larger plastic fragments were stored until use in another round of grinding and milling to generate MPs.

The choice of methods for degradation of plastics was based on different conditions under which these plastic types may become brittle and break, keeping in mind the actual degradation processes in the environment. Plastics such as PETE and PP become brittle when stored at very cold temperature, which could be further subjected to cryoblending for generating MPs. However, nylon from teabags is resistant to cold temperatures and can withstand high heat, leading to the use of microwave oven for breaking nylon tea bags. The processes employed also reflected the ease with which plastics could be broken down to generate MPs.

#### 2.2.4. Endotoxin Contamination Assessment

Although the plastic items chosen for generating MPs in the laboratory were thoroughly cleaned, they were previously used and exposed to household environmental conditions. Consequently, the lab-generated MPs underwent an endotoxin test, as endotoxins are abundant in the environment and can activate immune responses in cells, potentially obscuring the true effects of exposure to the MPs. Therefore, all lab-generated MPs were assessed for the presence of endotoxin using the ToxinSensor Chromogenic Limulus Amebocyte Lysate (LAL) Endotoxin Assay Kit (GenScript, Piscataway, NJ, USA) [59].

### 2.3. MP Characterization

#### 2.3.1. Raman Spectroscopy

The XploRA Plus confocal Raman microscope (Horiba Scientific, Piscataway, NJ, USA) served as the instrumental platform for the chemical identification of MPs, both those procured commercially and those synthesized in laboratory settings.

Water-suspended (2–3 µL) MPs were deposited on Calcium fluoride (CaF_2_) slides (Crystran, Poole, Dorset, UK) and visualized at 10×, 50×, and 100× magnification using brightfield illumination. Raman spectra of the deposited MPs were acquired at 532 nm laser excitation with 1 or 10 mW laser intensity depending on the size of the particles or fragments, using Labspec6 software, version 6.6.1.11 (Horiba Ltd., Piscataway, NJ, USA) [18]. The spectra were acquired with wavenumbers ranging from 500 to 3400 cm^−1^ for the analysis. The acquisition time and number of accumulations were adjusted to up to 20 s and ten accumulations, respectively, to obtain spectra of heightened signal to noise ratios [18]. The grating was set at 600 lines·mm^−1^. A slit size of 100 µm obtained the best resolution with the above settings. The FLAT correction was applied to remove background interference due to plastics’ auto fluorescence. The Raman spectra acquired with the Labspec6 software were compared with two publicly available spectral libraries of MPs—SLOPP and SLOPP-E libraries [60]—and a large fully verified spectral library of polymers (Bio-Rad Laboratories Inc., Hercules, CA, USA) using the KnowItAll software, version 24.1.65.0 [61].

#### 2.3.2. Scanning Electron Microscopy (SEM)

Suspensions of MPs at a concentration of 50 µg·mL^−1^ were prepared in ethanol in clean glass bottles by sonicating for 5 min using a Bransonic digital ultrasonic water bath at 40 kHz at room temperature (RT). About 2–3 µL of each MP suspension were deposited on an aluminum stub and dried for 20 min at RT. Since the MPs were not conducting materials, dried MPs on stubs were sputter coated with gold film to a thickness of 10 nm low vacuum sputter coater (Quorum, Q150T ES, Ashford, Kent, UK). The morphology and particle size of MPs were assessed by SEM analysis using Tescan VegaII XMU scanning electron microscopy (Brno, Czech Republic). The SEM images of at least 100 individual objects (particles, fibres, fragments) were used to determine the diameter, length, and width of the commercially purchased and lab-generated MPs using ImageJ (version 1.53g) software [62]. The results were plotted as histogram in SigmaPlot 15 (Systat Software, Inc., San Hose, CA, USA) and the standard deviation (SD) was calculated as the mean of 100 individual observations.

### 2.4. Cell Culture

Immortalized lung epithelial (FE1) cells derived from the transgenic Muta Mouse rodent model were used in the study. Cells were cultured in Dulbecco’s Modified Eagle’s Medium Nutrient Mixture (DMEM) made up of F12 HAM (1:1) culture media with 365 mg·L^−1^ L-Glutamine, (Life Technologies, Burlington, ON, Canada) supplemented with 2% fetal bovine serum (FBS, Life Technologies, Burlington, ON, Canada), 1 ng·mL^−1^ human epidermal growth factor (EGF, Life Technologies, Burlington, ON, Canada), 100 U·mL^−1^ penicillin G, and 100 µg·mL^−1^ streptomycin (Life Technologies, Burlington, Ontario, ON). Cells were maintained at 37 °C with 5% CO_2_.

### 2.5. Dynamic Light Scattering (DLS) and Electrophoretic Light Scattering (ELS) Characterization of MPs

MPs’ size distribution and polydispersity index (PDI, a measure of broadness of size distribution) in water and in the exposure medium were characterized by DLS method and the zeta potential of each MP type was characterized by ELS method using a Zetasizer nano ZSP (Malvern Panalytical, Westborough, MA, USA). Sonicated and suspended MPs in MilliQ water or in cell culture media using a Bransonic digital ultrasonic water bath at 40 kHz were diluted to a final concentration of 50 µg·mL^−1^, 100 µg·mL^−1^, or 1 mg·mL^−1^ (Appendix A details the sample preparation) and an aliquot of the suspensions was used for measuring the hydrodynamic diameter, polydispersity index, and the zeta potential.

### 2.6. Relative Cell Survival Assessment via Trypan Blue Exclusion Assay

For the relative cell survival assessment, cells were seeded at a concentration of 120,000 cells per well in 6-well plates and allowed to settle for 24 h in the incubator at 37 °C with 5% CO_2_. After 24 h, cells were treated with media only or media containing MPs at concentrations ranging from 6.25 µg·mL^−1^ to 50 µg·mL^−1^ or with 1 µg·mL^−1^ of lipopolysaccharide (LPS, *Escherichia coli* 055:B5, Sigma-Aldrich, Oakville, ON, Canada) for 24 h and 48 h. Media treated time-matched samples served as negative controls, 1% lysis buffer-treated samples served as experimental positive control for the Trypan blue assay, and LPS was used as a positive control for later experiments involving the assessment by enzyme-linked immunosorbent assay (ELISA). Percent relative cell survival compared with controls was determined using the Trypan blue exclusion assay as described in [63]. In brief, cells were trypsinized post-exposure by incubating with 150 µL of 0.25% Trypsin-EDTA (disodium ethylenediaminetetraacetic acid) and suspended in cell culture media. Suspended cells (10 µL) were combined with an equal volume of Trypan blue stain (Life Technologies, Burlington, ON, Canada) and incubated for 5–6 min at RT. Subsequently, live (unstained) and dead (blue-stained) cells were enumerated using a hemocytometer. Percent relative cell survival compared with media-treated time-matched controls were determined from the ratio of the number of live cells in a sample and the average number of live cells in the media controls (the percentage of relative survival (% relative survival = [Number of live cells·cm^−2^ of samples]/[number of live cells·cm^−2^ of the control] × 100) [63]. A Kruskal–Wallis one-way Analysis of Variance (ANOVA) on Ranks was conducted using SigmaPlot 15 to determine the significance between the exposed samples and matched media control. If the conditions for normality test (Shapiro–Wilk test) and equal variance passed, a one-way ANOVA with Dunnett’s post hoc was carried out to determine the significance between the exposed samples and matched media controls (three biological replicates) using *p* ≤ 0.05. The remaining cell suspensions were pelleted by centrifugation at 4 °C at 8000 RPM for 10 min. The cell supernatants and the cell pellets were collected separately and stored at −80 °C until the subsequent analysis.

### 2.7. Assessment of Pro-Inflammatory Mediators by ELISA

#### 2.7.1. Measurement of IL-1β and IL-6

The expression levels of IL-1β and IL-6 cytokines were measured individually using commercially available mouse enzyme-linked immunosorbent assay kit (Quantikine ELISA, BioTechne R&D Systems, Minneapolis, MN, USA), as described previously [64] and according to the manufacturer’s specifications. In brief, 50 μL of prediluted standards or 100 μL of the cell supernatants collected from FE1 cells exposed to media only, individual MPs, or LPS were loaded onto a microplate precoated with mouse IL-1β or IL-6 antibody and incubated for 2 h at RT on a shaker set at 300 rpm [64]. The unbound antibody was removed by washing the plates five times before adding 100 μL of mouse IL-1β or IL-6 conjugate to each well and incubating at RT for 2 h while rocking at 300 rpm. Plates were washed and 100 μL of substrate was added to each well and incubated with plates covered for 30 min. The reaction was stopped by adding 100 μL of the stop solution. The optical density of samples was determined at 450 nm using a SpectraMax 190 Microplate Reader (Molecular Devices, Sunnyvale, CA, USA) with the correction wavelength set at 540 or 570 nm. The detection limits for IL-6 and IL-1β as reported by the manufacturer are 1.8 and 4.8 pg·mL^−1^, respectively. A two-way ANOVA with a Dunnett’s post hoc was conducted using SigmaPlot 15 to determine the statistical significance between the exposed samples and matched media control (three biological replicates, two technical replicates) using *p* ≤ 0.05.

#### 2.7.2. Bio-Plex Pro Assay for the Assessment of Pro-Inflammatory Mediators

The levels of 23 pro-inflammatory cytokines in the cell culture supernatants collected following 48 h exposure to 100 nm PS, small PETE, PMMA MPs, medium, or LPS were determined using the Bio-Plex Pro Mouse Cytokine Group I 23-Plex (Bio-Rad Laboratory, Mississauga, ON, Canada). The specific cytokines assessed were interleukins- IL-1α, IL-1β, IL-2, IL-3, IL-4, IL-5, IL-6, IL-9, IL-10, IL-12 p40, IL-12 p70, IL-13, IL-17, Eotaxin, granulocyte colony-stimulating factor (G-CSF), granulocyte-macrophage colony-stimulating factor (GM-CSF), chemokine (C-X-C motif) ligand-1 (CXCL-1/KC), monocyte chemoattractant protein-1 (MCP-1) or C-C motif chemokine ligand-2 (CCL-2), macrophage inflammatory protein 1 alpha (MIP-1α/CCL-3), MIP-1β/CCL-4, INF-γ, RANTES/CCL-5, and TNF-α. The assay was performed according to the manufacturer’s instructions and as described previously [65]. In brief, anti-cytokine/chemokine antibody-conjugated magnetic beads were immobilized in each individual well of the 96-well plates using a vacuum manifold. Plates were washed twice and 50 μL of pre-diluted standards or 100 μL of the cell culture supernatants was added to each of the designated wells. Plates were incubated with the standards or samples at RT for 30 min at 350 rpm. Following 3 washes, plates were incubated for 30 min with 25 μL of pre-diluted multiplex detection antibody, washed 3 times, and incubated with 50 μL of pre-diluted streptavidin-conjugated Phycoerythrin for 10 min at RT at 350 rpm. Following final washes, plates were incubated with 125 μL of assay buffer in each well and were analyzed using the Bio-Plex 200 system (Bio-Rad, Hercules, CA, USA). The concentration of each chemokine/cytokine was obtained using the Bioplex Manager Software (version 6). For each assay, samples from three individual biological replicates were assessed and each sample was run twice (two technical replicates). A Student’s *t*-test using *p* ≤ 0.05 was conducted to determine the statistical significance.

### 2.8. Micronucleus Assay

For the micronucleus (MN) induction assay, FE1 cells were seeded into 96 well plates at the density of approximately 5 × 10^3^ cells in each well. Following the incubation for 24 h at 37 °C and 5% CO_2_, cells were treated with individual MPs at 0, 6.25, 12.5, 25, and 50 µg·mL^−1^ concentrations for 40 h. According to the most recent OECD guidance document for micronucleus induction assay [66], long term treatment can be accomplished by exposing logarithmically dividing cells to test articles for a duration of time that approximates 1.5 to 2 normal cell cycles. Since, FE1 cells have a doubling time of 18.7 ± 1.2 h [67], the 40 h duration of exposure was chosen to reflect 2 normal cell cycles. Methyl methanesulfonate (MMS) (Sigma–Aldrich, Oakville, ON, Canada), a known genotoxicant at a concentration of 500 µM, was used as a positive control. The cytotoxicity and MN induction were assessed via a flow cytometry-based method. The assay was performed using the In Vitro MicroFlow^®^ kit (Litron Laboratories, Rochester, NY, USA), according to the instructions provided with the kit [63]. In brief, following exposure, cells were washed with DMEM/F12 cell culture medium without phenol red. Freshly prepared Complete Nucleic Acid Dye A (50 µL, ethidium monoazide, or EMA) was added to each well to stain the apoptotic/necrotic cells. The plate was placed under a visible light source on ice for 30 min to allow the dye to bind DNA through photoactivation. At the end of the incubation period, 150 µL of cold 1X Buffer Solution was added to each well to stop the reaction. The supernatant was carefully removed. Cells were lysed and incubated in Complete Lysis Solution 1 containing Nucleic Acid Dye B and counting beads (6-micron fluorescent microspheres (Cat#C−16508, Life Technologies, Burlington, ON, Canada)) for 1 h at 37 °C to stain for healthy chromatin. Consequent to this step, cells were incubated with 100 µL of freshly prepared Completed Lysis Solution 2 containing Nucleic Acid Dye B at RT for 30 min. Cells were protected from light during the incubation periods. The data were acquired with the BD FACSymphony A1 cell analyzer with a High Throughput Sampler (BD Bioscience, San Jose, CA, USA) using FACS DIVA Software (Version 9.0.2). All dyes and physical parameters (forward and side scatter) were measured by the blue (excitation 488nm) laser. The FITC (530/30 nm detection) and the PerCP Cy5.5 (710/50 nm detection) channels were used to analyze fluorescent data. An analysis stop gate of 5 × 10^3^ EMA-negative nuclei was applied. The % MN induction and relative survival were determined according to the instructions provided by Litron Laboratories [68]. In brief, % MN induction was calculated using the count of MN event compared with the nucleated event. The results were expressed as relative fold change in MN induction in treated cells compared with average MN induction in matched media control [68]. The relative cell survival was calculated by comparing the nuclear density of the treated samples with the average nuclear density of the matched media control, where the nuclear density was calculated using the viable cells to bead ratio. The MN induction was considered positive if the change in MN induction fold change was statistically significant and ≥2 fold greater than the matched media controls. Statistical significance was determined using one-way ANOVA and Tukey’s post hoc multiple comparisons test using R studio version R-4.3.2. The final data represent 3 biological replicates per treatment condition.

### 2.9. Gene Expression Analysis

#### 2.9.1. RNA Extraction, Purification, and Integrity Analysis

Total RNA was isolated from cell pellets of FE1 cells exposed for 48 h to media control or individual MPs using TRIzol reagent (Cat#: 15596026, Invitrogen, Burlington, ON, Canada), and purified using RNeasy Plus Mini kits (Cat#: R2052, Zymo Research, Irvine, CA, USA) according to the manufacturer’s instructions. Total RNA concentration was measured using a NanoDrop one spectrophotometer (Thermofisher Scientific, Waltham, MA, USA). The RNA quality and integrity was assessed using an Agilent 4200 Tapestation System (Agilent Technologies, Inc., Santa Clara, CA, USA). Purified RNA was stored at −80 °C until analysis.

#### 2.9.2. Microarray Hybridization

The samples collected from cells exposed for 48 h to 0, 12.5. 25, and 50 µg·mL^−1^ of 100 nm PS, PETE, PMMA, or PP were used for analysis by microarrays. A random block design was applied [69,70]. Samples were hybridized according to the previously published method [71]. In brief, 200 ng of total RNA from each sample group as well as 200 ng of Universal Mouse Reference total RNA (UMRR, Agilent Technologies, Mississauga, ON, Canada) were reverse transcribed to synthesize double-stranded cDNA, which was then used to synthesize cyanine-labeled cRNAs. Equimolar amounts of reference cRNA (300 ng) and experimental cRNA (300 ng) were mixed and hybridized onto 8 × 60K Agilent SurePrint G3 Mouse Gene Expression v2 Microarray slides (Agilent Technologies, Inc., Santa Clara, CA, USA) in a hybridization oven for 17 h (65 °C, 10 rpm). Following this step, arrays were scanned on an Agilent G4900DA scanner. Gene expression data from the scanned images were extracted using Agilent Feature Extraction software version 11.01.1 software.

#### 2.9.3. Statistical Analysis of Microarray Data

Microarray data were normalized using LOcally WEighted Scatterplot Smoothing (LOWESS) regression modeling method [72] and the statistical significance of the differentially expressed genes (DEGs) was determined using MicroArray ANalysis of VAriance (MAANOVA) [73] in R statistical software (Version R-4.3.2). The exposure effect compared with the control was determined using Fs statistics [74]. For statistical significance, *p*-values were estimated using permutation methods, which were adjusted for multiple comparisons with false discovery rate test corrections [75]. All microarray data have been deposited in the NCBI gene expression omnibus database and can be accessed via the accession number GSE269348.

#### 2.9.4. Pathway Enrichment Analysis

Ingenuity Pathway Analysis (IPA) tool (QIAGEN, Redwood City, CA, USA) was used to identify enriched canonical pathways, upstream regulators, diseases, and functions associated with DEGs. For PETE-exposed groups, genes exhibiting an FDR adjusted *p*-value of ≤0.05 and absolute fold change (FC) of 1.5 were considered as DEGs. For 100 nm PS, PMMA, and PP MPs, genes showing unadjusted *p*-value ≤0.05 and absolute FC of 1.5 were used in the analysis, as there were very few DEGs by the FDR-adjusted *p*-value cut off. Pathways exhibiting a Fisher’s exact *p*-value ≤ 0.05, absolute Z-score ≥ 2, and a minimum of three genes were considered enriched and included in the interpretation of results. The z-score is calculated in IPA tool to measure how closely the expression of DEGs in a pathway corresponds to the literature-derived gene expression patterns indicative of pathway activation or deactivation. Pathways with a Z-score ≥ 2 were considered activated and Z-score ≤ −2 were considered deactivated.

The pathway and biological process enrichment analysis was also carried out using the Metascape Analysis tools, version 3.5 [76]. For ontology source, the Gene Ontology (GO) Biological Processes were applied and all the genes in the genome were used. Terms with a *p*-value < 0.01, a minimum count of three genes, and an enrichment factor > 1.5 (the ratio between the observed counts and the counts expected by chance) were grouped into clusters based on their membership similarities, where *p*-values were calculated based on the cumulative hypergeometric distribution [75] and *q*-values were calculated using the Benjamini–Hochberg procedure to account for multiple testing [77]. Kappa scores were used as the similarity metric for hierarchical clustering of the enriched terms [78] and subtrees with a similarity of >0.3 were considered a cluster.

### 2.10. Detecting MPs in Cells

#### 2.10.1. Immunofluorescence Labeling

FE1 cells were seeded on coverslips at a concentration of 60,000 cells per well in a 6-well plate and incubated at 37 °C with 5% CO_2_ for 24 h to settle. Next day, cells were treated with 50 µg·mL^−1^ of small PETE or with medium only for 24 h. Following exposure, cells were washed twice with cell culture media and once with the warm Phosphate Buffered Saline (PBS). Cells were fixed by incubating in freshly prepared 10% Formalin (Sigma-Aldrich, Oakville, ON, Canada) in PBS for 10 min at RT, permeabilized by treating with 1% Triton-X in PBS with 0.1% Tween−20 (PBST, Thermofisher, Burlington, ON, Canada) before staining the cells.

After two washes with PBS, a blocking step was applied to prevent non-specific binding with 1% Bovine Serum Albumin (Sigma-Aldrich, Oakville, ON, Canada) in PBST. Cells were stained with Phalloidin Fluor 594 reagent (Abcam, Toronto, ON, Canada) in 1% BSA in PBS for 1 h, at RT in dark to mark actin. Following three washes with PBS, the coverslips were mounted on slides using ProLong glass antifade with NucBlue mounting media containing nuclear counterstain Hoechst (Thermofisher, Burlington, ON, Canada) to mark DNA.

#### 2.10.2. Immunofluorescence and Enhanced Darkfield Hyperspectral Imaging (EDF-HSI)

The EDF-HSI system (CytoViva, Inc. Auburn, AL, USA) and its principles for detection of nanoscale materials have been described previously [18]. In brief, hyperspectral data are collected as hypercubes in the field of view per scan acquisition and processed using ENVI v. 4.8 software (Harris Geospatial Solutions). The optical spectrum from 400 to 1000 nm for a given particle is recorded at a single pixel level, which is used to construct a spectral library for that particle or, in this case, an individual MP. The resulting reference library is then used to detect the presence of that particular MP in a complex matrix (e.g., in cells after exposure) by correlational spectral angle mapping using a spectral classification algorithm. The identity of the particles is confirmed when the spectral angles between the reference spectral library and sample spectra are ≤0.1 radian. In the current study, EDF-HSI was used to create spectral libraries of lab-generated PETE MPs for the purpose of tracking them in FE1 cells post-exposure. Hyperspectral data were collected in the partial field of view (501 lines per image) at 0.25 s exposure per scan acquisition.

The orientation and localization of small PETE within the actin cytoskeleton of FE1 cells, was determined by capturing images of small PETE MPs in 3D space. A stack of 121 equally spaced (50 nm spacing) images of the sample at 100× magnifications with spatial resolution of 64.5 was acquired and stored using the CytoViva EDF Microscope equipped with a piezo-driven Z-axis stage, a multi-mode fluorescence module, and CytoViva’s image acquisition and control software. The images of the PETE MPs were captured under dark field illumination using direct light from a tungsten–halogen light source, while the images of the actin cytoskeleton and nuclei were captured with light from fluorescence module passing through a triple-pass filter. The z resolution was set to 200 nm. The stack of images of the PETE MPs and the surrounding cell components were collapsed using the CytoViva 3D Image Analysis tool in ImageJ plugin via interpolation and deconvolution, respectively, to obtain 3D renderings. The 3D renderings were visualized using the 3D viewer in the ImageJ plugin.

## 3. Results

### 3.1. MPs Characterization

#### 3.1.1. Lab Production of MPs

Recognizing the limitations of commercial MPs to replicate real-world environmental exposures and being faced with the difficulties in obtaining sufficient quantities of MPs from authentic environmental samples for toxicological evaluation, it was decided to generate MPs from plastic items representing commonly encountered plastic debris found in various environmental settings. The selection of plastic items targeted for MP generation was based on the MP types detected in outdoor air samples [18]. The results underscored the necessity of tailoring the method for MP generation based on the plastic type, requiring the optimization of multiple methods. The manual methods employed were labor-intensive and yielded MPs exhibiting diverse sizes and shapes, potentially leading to batch-to-batch discrepancies, thereby mandating characterization for each produced batch. The resulting MPs consisted of particles, fragments, and fibres, and exhibited different suspension properties in cell culture media (Table 1). It was found that not all plastic types are amenable for MP generation applying the methods optimized in this study. Moreover, the yield from each plastic type varied. In contrast, commercial MPs were mostly of uniform size, with some variation in PMMA, LDPE, and HDPE sizes, uniform shapes (except for LDPE), and were homogeneously suspended.

#### 3.1.2. Chemical Identity of MPs

Raman spectra of individual MPs accurately identified the commercial plastic polymers as PS, PMMA, LDPE, and HDPE, and the lab-generated MPs as nylon, PETE, and PP, in the KnowItAll database and SLOPP and SLOPPE plastic polymer databases, with high accuracy. There were no other chemicals detected, attesting to the purity of the MPs investigated. The brightfield images and their corresponding Raman spectra for all MPs are presented in Figure 2A,B. Appendix A show the sample vs reference polymer spectra from the database to which the sample spectrum was matched and specify the hit quality index (HQI).

#### 3.1.3. SEM

The SEM images of individual MPs and the size distribution histogram plots are presented in Figure 3 and Figure 4. The morphologies of the commercial MPs were spherical, except for LDPE, which exhibited irregular shapes. The diameter of commercial MPs as measured by the SEM was subtly different compared with the manufacturer reported diameter; the average SEM diameter of 60 nm PS, 100 nm PS, 300 nm PS, PMMA, and HDPE were 56.2 ± 1.6 nm, 97.67 ± 0.09 nm, 245.1 ± 5.5 nm, 3.7 ± 0.2 µm, and 4.3 ± 0.3 µm, respectively. The average length and width of LDPE were 3.1 ± 0.1 µm and 1.6 ± 0.09 µm, respectively. While the size distribution observed for the PS MPs was very tight, the size distribution was broad for PMMA, LDPE, and HDPE, and ranged from 1 to 40 µm.

The SEM images of lab-generated nylon, PETE, and PP MPs are presented in Figure 4. Unlike the commercial MPs, the lab-generated MPs were fragmented and irregular in shape, except for nylon MPs fractionated from bulk nylon polyamide powder which were spherical. All lab-generated MPs had a broad size distribution ranging from nanometer to 2.7 µm, which was the maximum size cut-off set for all lab-generated MPs. There were two distinct size populations for both nylon MP types. The nylon MPs prepared from bulk nylon polyamide powder exhibited diameters in the range of 0.26 ± 0.01 µm and 1.54 ± 0.01 µm. The nylon MPs extracted from tea bags exhibited two sizes; the length and width of the first size population were 0.27 ± 0.01 µm and 0.14 ± 0.01 µm, respectively, and the length and width of the second size population were 1.54 ± 0.01 µm and 0.54 ± 0.01 µm, respectively. The PETE MPs were prepared in two sizes. The small PETE were 0.81 ± 0.03 µm and 0.24 ± 0.01 µm in size, respectively, whereas the large PETE were 1.47 ± 0.06 µm and 0.35 ± 0.01 µm size, respectively. The PP were 1.01 ± 0.03 µm and 0.83 ± 0.03 µm in size. The aspect ratios of the lab-generated MPs ranged from 0.77 µm to 3.05 µm.

#### 3.1.4. DLS and ELS Analysis of MPs

The results of the DLS analyses of all MPs are presented in Table 1. The average hydrodynamic sizes of PS and LDPE in MilliQ water or cell culture media suspensions were in the nanometer range (62.25 ± 0.36 to 342.34 ± 82.5 nm). The average hydrodynamic sizes of PMMA and HDPE fell within the micrometer range, spanning from 1.09 to 3.65 µm. Notably, for PMMA, a smaller size fraction was also identified, with 2.4% of the suspension exhibiting a size of 29.9 nm in MilliQ water, and 3.5% displaying a size of 38.6 nm in cell culture media. All three PS NPs exhibited mono dispersion in MilliQ water and in cell culture media, with a PDI of ≤0.1. The other commercial MPs showed heterogeneous dispersion in MilliQ water and in the cell culture medium, with the PDI index ranging from 0.48 to 0.78. The zeta potential measurements of the commercial MPs ranged from −39.4 to −50.38 mV in MilliQ water and from −5.03 to −15.72 mV in the cell culture media. The hydrodynamic sizes of lab-generated MPs in MilliQ water and in the cell culture media ranged from ~0.2 to ~1.8 µm. The nylon MPs obtained from bulk polyamide powder displayed a hydrodynamic size of approximately 4.5 µm in both MilliQ water and cell culture media. Conversely, nylon MPs extracted from tea bags exhibited smaller sizes, approximately 0.7 µm and 0.2 µm in MilliQ water and cell culture media, respectively. In MilliQ water, over 97% of MPs were around 0.7 µm in size, while in cell culture media, 62% of MPs measured approximately 0.7 µm, and 38% were ≤46.1 nm. The hydrodynamic sizes of small PETE MPs were approximately 0.4 µm in MilliQ water and ~0.7 µm in cell culture media. For large PETE MPs, two size populations of around 1.1 µm in MilliQ water and 0.7 µm in cell culture media were observed. For small and large PETE, over 99% of MPs measured approximately 0.4 µm in MilliQ water, whereas less than 90% of MPs were smaller than 2.1 µm in cell culture media. For the PP, the hydrodynamic sizes were about 0.2 in MilliQ water and 1.8 µm in cell culture media. The PDI index was above 0.1, indicating hetero dispersion in MilliQ water and in cell culture media. The zeta potential of the lab-generated MPs spanned from −31.18 to −52.28 mV and −7.08 to −17.4 mV in MilliQ water and in cell culture media, respectively.

#### 3.1.5. Endotoxin Assessment Results

The results from the endotoxin assay are presented in Appendix A. The endotoxin levels in lab-generated nylon, PETE, and PP MPs were negligible (<0.01 EU·mL^−1^).

### 3.2. Toxicity Assessment of MPs

#### 3.2.1. Relative Cell Survival

##### Commercial MPs

All commercial MPs induced significant decreases in % relative survival compared with the media-treated controls, with some exhibiting clear dose-dependency (Figure 5A). At 24 h post-exposure, the 60 nm PS treatment resulted in cell growth at all concentrations. The 100 nm PS and the 300 nm PS did not show a decrease in % relative survival at any concentration. After 48 h post-exposure to 60 nm PS, at a concentration of 50 µg·mL^−1^, the % relative survival displayed a statistically significant decrease, measuring 66.3%. However, the decreases were not statistically significant at the lower concentrations of 12.5 µg·mL^−1^ and 25 µg·mL^−1^, where the % relative survivals remained relatively high at 78.6% and 92.2%, respectively. This suggests a dose-dependent relationship, with higher concentrations correlating with more pronounced decreases in % relative survival. Similarly, at 48 h, the 100 nm PS showed 83.6%, 77.6%, and 71.1% survival at 12.5, 25, or 50 µg·mL^−1^ concentrations, respectively. However, the response at the lowest dose was not significant. The 300 nm PS induced a statistically significant decrease in % relative survival at 50 µg·mL^−1^ (83.8%) concentrations.

The treatments involving PMMA, LDPE, and HDPE consistently resulted in decreased % relative survivals at both time points. For PMMA, at 24 h post-exposure, % relative survival decreased to 80.1%, 65.8%, and 61.1% when exposed to concentrations of 12.5, 25, or 50 µg·mL^−1^, respectively. At 48 h post-exposure, the % relative survival further decreased to 76.6%, 63.6%, and 50.8% for the same concentrations. Post-LDPE treatment, the % relative survival reduced to 90.9%, 84.5%, and 81.1% at 24 h and to 96.2%, 91.8%, and 85.8% at 48 h at the three concentrations, respectively. The HDPE-treated group showed 82.3%, 82.1%, and 63.7% survival at 24 h and 81.6%, 84.3%, and 72.5% survival at 48 h, at 12.5, 25, or 50 µg·mL^−1^ concentrations, respectively. The decrease in relative survival was statistically significant for all concentrations of PMMA at 48 h post-exposure, for high concentration groups of LDPE and for all concentrations of HDPE for both time points.

##### Lab-Generated MPs

A decrease in % relative survival was observed for both nylon MP types at all concentrations and timepoints (Figure 5B). At 24 h post-exposure to nylon from bulk polyamide powder, the % relative survival reduced to 76.3%, 73.1%, and 61.6%, and to 85.1%, 73.8%, and 60.6% at 48 h at the three concentrations. In cells treated with nylon MPs from tea bags, the % relative survivals decreased to 93.0%, 93.5%, and 79.8%, at 24 h, and to 96.0%, 92.1%, and 78.1% at 48 h post-exposure to 12.5, 25, or 50 µg·mL^−1^ concentrations, respectively. The results were significant for all concentrations for the nylon MPs prepared from bulk powder and at high concentrations of nylon MPs from tea bags.

At 24 h post-exposure, neither small nor large PETE MPs led to a reduction in % relative survival at any concentration. However, at 48 h post-exposure, reductions in % relative survival were observed. Specifically, exposure to small PETE MPs resulted in reductions to 82.9% and 76.8% at concentrations of 25 and 50 µg·mL^−1^, respectively. Likewise, exposure to large PETE MPs led to reductions to 92.5% and 84.9% at the same concentrations. At 24 h post-exposure, % relative survival decreased to 79.5%, 82.1%, and 75.8% when exposed to PP MPs at concentrations of 12.5, 25, or 50 µg·mL^−1^, respectively. By 48 h post-exposure, % relative survival further decreased to 78.1%, 72.8%, and 62.4% for the same concentrations. Among the lab-generated MPs tested, PP MPS exhibited the highest toxicity, while large PETE demonstrated the lowest toxicity. These findings underscore the considerable toxicity of PP MPs, evidenced by significant decreases in % relative survival over time and across various concentrations, contrasting with the comparatively lesser impact observed with large PETE.

#### 3.2.2. Pro-Inflammatory Cytokines Expression

To determine if cellular exposure to individual MPs results in pro-inflammatory response, the levels of IL-6 and IL1-β protein were assessed in cell culture medium post-48 h MP exposure. IL-6 and IL1-β were specifically targeted due to their well-established roles in inflammation. These cytokines play crucial roles in the body’s immune response, particularly in the initiation and regulation of inflammatory processes. Among the six commercial MPs tested, 100 nm PS and PMMA induced subtle increases in IL-6 expression at different concentrations; however, the results were not significant (Figure 6A). None of the commercial MPs induced any detectable increase in the expression of IL-1 β at this time point (Figure 6A). Among the lab-generated MPs tested, exposure to small PETE and large PETE resulted in notable increases in IL-6 levels. Specifically, there was a 2.5-, 131-, and 177-fold and a 0.8-, 1.3-, and 2.2-fold increase in IL-6 levels observed after exposure to 12.5, 25, and 50 µg·mL^−1^ concentrations of small PETE or large PETE, respectively (Figure 6B). The results were significant for 25 and 50 µg·mL^−1^ concentrations for small PETE and a 50 µg·mL^−1^ concentration of large PETE (Figure 6B). However, these MPs did not induce changes in IL-1 β protein levels. The concentration of cytokines in the supernatants following MP exposure are presented in Appendix A.

Since some MPs induced the expression of IL-6, a 23-cytokine Bioplex assay was used to assess potential changes in the expression of other pro-inflammatory cytokines and chemokines. A select set of three individual MP types were chosen for the Bioplex analysis. The results are presented in Figure 7. In comparison with LPS treatment, which induced the expression of 14 of 23 cytokines tested, treatment with MPs resulted in a subtle increase in the expression of a very small number of pro-inflammatory mediators. The 100 nm PS induced the altered expression (approximately 1.3-fold) of four, three, and three cytokines at 12.5, 25, and 50 µg·mL^−1^ concentrations, respectively, compared with the untreated controls at 48 h post-exposure. The small PETE induced a larger than 1.3-fold increase in the expression of 8 cytokines at each of the three concentrations tested; these cytokines included, IL-3, IL-5, IL-6, IL-12 (p40), IL-13, Eotaxin, and RANTES. PMMA induced the altered expression of three, four, and four cytokines at 12.5, 25, and 50 µg·mL^−1^ concentrations at the same post-exposure timepoint.

#### 3.2.3. MN Induction

Cells were exposed to various concentrations of commercial and lab-generated MPs for 40 h. The % relative cell survival and fold-increase in MN induction were determined. The results are presented in Figure 8. The 60 nm or the 300 nm PS MPs did not induce MN formation at any concentration. The 100 nm PS induced a dose-dependent increase of 1.8-, 2.2-, 3-, and 3.5-fold at 6.25, 12.5, 25, and 50 µg·mL^−1^ concentrations, respectively. The changes were statistically significant at 25 and 50 µg·mL^−1^ concentrations. Although the results for both PMMA and HDPE were not statistically significant, a dose-dependent trend was observed, prompting their inclusion in the interpretation of the results. PMMA induced increases of 0.8-, 1.3-, 2.0-, and 1.7-fold at concentrations of 6.25, 12.5, 25, and 50 µg·mL^−1^ respectively, while HDPE induced increases of 2.6-, 2.1-, 1.8-, and 2.4-fold at the same concentrations, respectively. The nylon MPs from bulk powder showed a 2.3-fold and 2.2-fold increase in MN at 12.5 and 50 µg·mL^−1^ concentrations, respectively. The nylon MPs from tea bags did not have any impact on MN formation. The small PETE MPs induced dose-dependent increases of 1.6-, 2.0-, 2.6-, and 3.0-fold in MN at 6.25, 12.5, 25, and 50 µg·mL^−1^ concentrations, respectively. The large PETE MPs induced 2.3-fold MN induction only at the high concentration of 50 µg·mL^−1^. The changes were statistically significant at the highest concentration for the large PETE MPs and at 25 and 50 µg·mL^−1^ concentrations for the small PETE MPs. The % relative cell survival was ≥40% compared with the matched media controls at all concentrations except for PMMA and HDPE MPs at the highest concentration, at which 30.9% and 21.9% survival was recorded, respectively. The % relative survival calculated from the Microflow assay and from the Trypan blue assay were similar.

#### 3.2.4. Gene Expression Analysis

Based on the results obtained for other endpoints, four individual MPs—100 nm PS, PMMA, small PETE, and PP showing varying levels of toxicity—were selected for the microarray analysis. Because the cellular cytotoxicity response at 24 h post-exposure was inconsistent across these MP types, samples were exclusively collected at the 48 h mark to ensure a more reliable assessment of gene expression changes using the microarray technique. Of the four, only small PETE induced changes in the expression of a significant number of DEGs (FDR adjusted *p*-value ≤ 0.05 and absolute FC of 1.5). The list of top 10 upregulated and top 10 downregulated DEGs from this treatment are presented in Appendix A. On the contrary, the 100 nm PS, PMMA, and PP MPs induced fewer than five DEGs and thus, the *p*-value cut off for this group of MPs was relaxed to FDR unadjusted *p*-value ≤ 0.05 with an absolute FC cut off of 1.5. These genes are listed in Appendix A and the number of up- and downregulated genes for all concentrations are summarized in Appendix A.

In all, small PETE induced 141 (upregulated), 170 (163 upregulated and 7 downregulated), and 450 (310 upregulated and 140 downregulated) DEGs at 12.5, 25, and 50 µg·mL^−1^ concentrations, respectively, (Figure 9A). Of these DEGs, 141 (138 upregulated and 3 downregulated) were common across the PETE treatment groups (Figure 9B). The top ten DEGs are listed along with their expression levels in Figure 9C.

The 100 nm PS induced 3 (1 upregulated and 2 downregulated), 53 (22 upregulated and 31 files downregulated), and 20 (8 upregulated and 12 downregulated) DEGs at the 12.5, 25, and 50 µg·mL^−1^ concentrations, respectively (Appendix A). PMMA induced 17 (5 upregulated and 12 down regulated), 82 (49 upregulated and 33 down regulated), and 10 (4 upregulated and 6 down regulated) DEGs at the 12.5, 25, and 50 µg·mL^−1^ concentrations, respectively (Appendix A).

PP MPs induced 75 (55 upregulated and 20 down regulated), 153 (105 upregulated and 48 down regulated), and 37 (17 upregulated and 20 down regulated) genes significantly at the 12.5, 25, and 50 µg·mL^−1^ concentrations, respectively (Appendix A). The Venn analysis showed very few DEGs in common across the concentration groups within a specific MP type or across the MPs investigated. Common DEGs across the concentration groups within a specific MP type are summarized in Appendix A.

The pathway analysis using the IPA tool revealed that the upregulated and downregulated DEGs in small PETE treatment groups perturbed 33, 34, and 58 canonical pathways at 12.5, 25, and 50 µg·mL^−1^ concentrations, respectively (Figure 10A), most of which were activated. A majority of these canonical pathways were associated with the immune response, i.e., cytokine signaling (interferon signaling, ISG15 antiviral mechanism interleukin-1 family signaling), cellular immune system (Systemic Lupus Erythematosus in B Cell Signaling Pathway, role of PKR in interferon induction and antiviral response and T cell receptor signaling), adaptive immune system (Class I MHC-mediated antigen processing and presentation), and innate immune system (DDX58/IFIH1-mediated induction of interferon-alpha/beta and cytosolic sensors of pathogen-associated DNA) responses (Figure 10B). A perturbation in pathways associated with inflammation (e.g., neuroinflammation, hypercytokinemia/hyperchemokinemia, pathogen-induced cytokine storm signaling pathway), cellular stress and injury (ISGylation Signaling Pathway, Cachexia Signaling Pathway, Necroptosis Signaling Pathway), apoptosis (Necroptosis Signaling Pathway, Death Receptor Signaling and Retinoic Acid-Mediated Apoptosis Signaling), cell cycle (Mitotic G1 phase and G1/S transition and regulation of mitotic cell cycle), and DNA replication (DNA replication pre-initiation) was also observed at the highest concentration of 50 µg·mL^−1^ (Figure 10C). In the treatment groups of 100 nm PS, PMMA, and PP, no significant pathway perturbations were noted for 100 nm PS-treated groups (absolute Z score ≥ 2). PMMA-induced pathway perturbations did not show dose-dependency, with seven canonical pathways activated at the 25 µg·mL^−1^ concentration and none at the 12.5 or 50 µg·mL^−1^ concentrations. PP MPs induced the activation of fourteen and four canonical pathways and the deactivation of three and one canonical pathways at 25 and 50 µg·mL^−1^ concentrations, respectively (Appendix A). The pathways associated with cell cycle (Mitotic Prometaphase, Cell Cycle Checkpoints, and Mitotic Metaphase and Anaphase) were activated at 25 µg·mL^−1^ of PMMA and at 50 µg·mL^−1^ of PP (Appendix A). In addition, the inactivation of apoptotic pathways (p53 Signaling, PD-L1 cancer immunotherapy pathway, and DNA damage-induced 14-3-3σ signaling) and the activation of pro-inflammatory pathways (Docosahexaenoic Acid (DHA) Signaling, Systemic Lupus Erythematosus in B Cell Signaling Pathway, Dendritic Cell Maturation, and NAD Signaling Pathway) was noted for the PP group (Appendix A).

A parallel analysis of all DEGs for the enrichment of biological processes and functions revealed (Appendix A) an association with the GO terms of the immune response at all concentrations of small PETE and an association with the processes such as angiogenesis, inflammation, cell population proliferation, extracellular matrix organization, response to oxidative stress, and apoptosis processes at the high concentration of 50 µg·mL^−1^.

The DEGs from the other MP treatment groups were also associated with the GO processes, such as cell–matrix adhesion, angiogenesis, the positive regulation of the cell cycle process, and carbohydrate metabolic processes in cells treated with 100 nm PS; immune response, cell proliferation, inflammation, and apoptosis in cells treated with PMMA; and nucleosome assembly, the regulation of small molecule metabolic processes, skeletal system development, the negative regulation of RNA splicing, negative regulation of DNA recombination, negative regulation of protein catabolic process, and the regulation of apoptotic signaling pathways in the PP-treated group (Appendix A).

### 3.3. Cellular Uptake of MPs

FE1 cells exposed to media or small PETE were visualized using epifluorescence imaging and EDF-HSI to detect the cellular uptake of small PETE at 24 h and 48 h post-exposure. The distribution of actin (pseudo colored in red), nucleus (pseudo colored in blue), and MP (bright white objects) in cells are shown in Appendix A. The field of view where bright white objects were detected were used to acquire EDF-HSI. The results show small PETE in several cells, as in Appendix A. Any bright white objects detected in the media-treated controls did not map to the spectra of small PETE. Figure 11 shows the 2D and 3D view of a small PETE particle or fragment interacting or piercing the actin cytoskeleton. Disorganization of actin filaments was also observed at 48 h post-exposure (Appendix A).

## 4. Discussion

This study established lab-scale methods for generating MPs for toxicological studies and investigated six commercial and five lab-generated MPs for their potential to induce toxicity in vitro. The tested MP types reflected the plastics polymers used in products and plastic litter found in the environment. For example, PS (Styrofoam) are used in construction, household appliances, arts and crafts, and in food packaging. LDPE is a transparent plastic with low crystallinity (50–60%), from which plastic bags are made. HDPE is rigid compared with LDPE and because of its high tensile strength and resistance to heat, it is used for making fibres (e.g., ropes, nets, fabrics), packaging material, plastic pipes, tubing, and play structures. PMMA is a transparent engineered plastic used for making acrylic glass and molding compounds such as dials, rulers, lenses, and plastic optical fibers. Different grades of nylon are used in a plethora of products used in the automotive industry and the textile industry, as well as in the health and sports industry. Nylon is also used to make tea bags [79]. PETE is the third most used plastic in packaging and is commonly found in water and soft drink bottles, plastic films, food packaging, and in cosmetics [80]. Finally, PP is a commodity plastic used to make fibers and fabrics, film, and sports equipment, as well as medical and laboratory tools [81]. However, it is important to note that commercial MPs, because of their uniform size and shapes, do not reflect what is found in the environment and, on the other hand, the lab-generated MPs in this study were clean and were of non-uniform sizes and shapes, but they represent only a fraction of what is found in the environment.

As previously mentioned, currently available MPs for purchase do not accurately reflect environmental exposure. Furthermore, standardized laboratory techniques for producing relevant MPs are lacking. Although the methods employed in this study yielded enough MPs for toxicological analysis, they are labor-intensive. The lab-generated MPs replicated the size and shape heterogeneity of respirable outdoor air MPs; however, they lacked the environmental contaminants found adsorbed on MPs’ surfaces in ambient air, such as carbon black, chemicals, and pathogens. Nonetheless, this study generated data to understand the toxicity of the raw plastic MPs without the additives, which was the primary aim of this study.

All MPs investigated in the present study induced cytotoxicity in exposed lung epithelial cells. Several studies in the literature have reported reduced cell viability following exposure to PVC or PMMA in lung fibroblast cells (IMR 90) [82,83] after exposure to PS in A549 cells [82] and in human MCF-7 breast cancer cell lines and MDA-MB-231 cells after exposure to PP MPs [84]. Size-dependent and dispersing-agent-dependent uptake and accumulation of PETE NPs, abnormalities in hatching rate and heart rate, and the generation of oxidative stress have been documented in Zebra fish [85]. PETE of a size range of 122–221 nm was shown to cause decreased mitochondrial membrane potential of A549 cells at a concentration of 49.1 µg·mL^−1^ [22]. PETE of this size range and concentration was also shown to induce oxidative stress and ROS production in A549 cells [22,86]. These studies suggest that different MPs can induce cytotoxicity and toxicity in general in different cells in culture and in different model systems.

At 48 h post-exposure, most MPs induced a response in the context of various endpoints tested and thus the 48 h timepoint was chosen for the gene expression analysis. Although subtle, the expression of many cytokines and chemokines was found to have altered after MP exposure in the present study, with IL-6, CCL5, and Eotaxin expression showing a dose-response. Both CCL5 and Eotaxin are associated with eosinophil chemotaxis, and an increased number of eosinophils are observed in allergic and asthmatic conditions [87]. Elevated levels of both IL-6 and CCL5 are also found in various inflammatory conditions. The commercial 100 nm PS, PMMA, and the lab-generated PETE MPs induced an increased expression of pro-inflammatory cytokines in a potency order of small PETE > 100 nm PS > PMMA > large PETE, suggesting potential for MPs to induce allergic reactions and, consequently, inflammation.

Of all the MPs tested for their potential to induce genotoxicity, only 100 nm PS and small PETE induced MN formation, with small PETE showing a clear dose-response and a higher magnitude of MN induction. To understand the underlying mechanisms of toxicity of MPs, whole genome gene expression analysis was conducted following 48 h exposure to 100 nm PS, PMMA, small PETE, or PP MPs (the types that showed positive response as measured by the different endpoints) using microarrays. Only small PETE induced a significantly altered expression of a large number of genes, in a dose-dependent manner.

Small PETE exposure resulted in a robust activation of the interferon signaling pathway involving 2′-5′ oligoadenylate synthetase 1A, ISG15 ubiquitin-like modifier, 2′-5′ oligoadenylate synthetase-like 2, 2′-5′ oligoadenylate synthetase 2, receptor transporter protein 4, interferon regulatory factor 7, ubiquitin specific peptidase 18, ubiquitin specific peptidase 18, chemokine (C-X-C motif) ligand 10, lymphocyte antigen 6 complex, locus C1, interferon-induced protein with tetratricopeptide repeats 1, MX dynamin-like GTPase, signal transducer and activator of transcription 1 and interferon, and alpha-inducible protein 27 like 2A, all of which were upregulated by more than 20-fold. The 2′-5′ oligoadenylate synthetase (OAS) 1A, belonging to the family of molecular innate immune sensors was upregulated by more than 130-fold. The interferon pathway and interferon-stimulated genes (ISGs) sense cytosolic double-stranded RNA, which is a potent indicator of viral infection [88,89]. However, in the present study, the lab-generated MPs did not show endotoxin contamination, suggesting that cells may be perceiving MPs as PAMPs (Pathogen Associated Molecular Patterns).

MPs have been suggested to activate the innate immune system upon interaction with cell membrane and/or cellular internalization, potentially leading to the activation of inflammasome, a type of immune response mounted following PAMP signaling (reviewed in Alijagic et al.) [90]. Amino-modified PS MPs ≤ 100 nm have been shown to induce NLRP3 (tripartite protein consisting of a central nucleotide-binding and oligomerization domain (NOD), a C-terminal leucine-rich repeat (LRR) domain, and an amino-terminal pyrin domain (PYD)) activation, a component of inflammasome in human primary macrophages [91], in THP1 cells [54], and in mouse lung (MLE-12) cells [92]. In another study, non-functionalized PS MPs were shown to induce the gene and protein expression of NLRP3 in mouse hepatocytes (AML12) [93]. Oral exposure to non-functionalized PS for 90 days was also shown to induce inflammation and oxidative stress in rats by NLRP3 activation [94,95]. Inflammasome has also been documented following exposure to engineered nanomaterials [96,97,98,99,100]. For engineered nanomaterials that are insoluble, bio persistent, or that exhibit a high aspect ratio, interaction with lung cells and cellular components resulting in an activation of PAMP and Danger Associated Molecular Patterns (DAMP) signaling is suggested to be a key molecular event in the manifestation of lung inflammation, lung injury, and the transition to lung fibrosis [101]. Although various MPs have been shown to induce inflammation and oxidative stress in vitro and in vivo by NLRP3 activation, many MPs have also been shown to induce toxicity in an NLRP3-independent manner. In a study of eight MPs including amino-functionalized and non-functionalized PS MPs, PE MPs, PETE MPs, polyester fibre, polyacrylonitrile fibers, and nylon fibres by Busch et al., the increased expression of IL-1β and the activation of the NLRP3 pathway were observed in THP-1 cells only for amino-functionalized PS [54]. It was shown that non-functionalized PS MPs, PE MPs, PETE MPs, polyester fibre, polyacrylonitrile fibers, and nylon fibres induce the secretion of pro-inflammatory IL-8 without the secretion of IL-1β. In addition to activating interferon signaling pathways, small PETE also induced several DEGs associated with inflammation, DNA damage, and cell injury. Thus, the potential to induce cytotoxicity, the expression of pro-inflammatory mediators, and MN induction, together with a robust alteration in the expression of a large number of DEGs, suggests that small PETEs can induce injury at the tissue level, an adverse outcome preceding disease and tissue dysfunction [102], and most importantly, in the context of inhalation exposure, lung fibrosis. There were a few genes that were downregulated following exposure to small PETE; however, the fold changes hovered around approximately 2-fold, with only a couple of them showing dose response but not specifically associated with any of the enriched pathways.

The other three MPs—100 nm PS, PMMA, and PP—did not induce statistically significant alterations in gene expression. An FDR unadjusted *p*-value cut-off of 0.05 resulted in a few genes exhibiting marginal fold-change in the expression of different genes than the ones observed following small PETE exposure. These results suggest that small PETE MPs are the most potent MPs of all the MPs tested in this study.

The chemical identification of MPs using the Raman method revealed PETE as the main chemical species in the disposable water bottles used for generating small PETE MPs in this study. While Phthalates and Bisphenol A (BPA) from plastics has been shown to induce immune response including interferon signaling [103], we did not detect BPA or any other chemical groups in PETE from water bottles that were used to generate MPs. Also, larger PETE MPs prepared from the same bottles, differing only in size, did not induce the responses observed following small PETE, suggesting that the non-homogenous size and shapes of small PETE, which included particles and fragments of irregular dimensions, may have allowed for unique cellular interactions with MPs. Smaller sizes impacting their distribution (aggregation/agglomeration) in the culture medium, leading to effective cellular internalization, may have triggered the PAMP-like signaling [101]. Small PETE was internalized by the cells as observed by the EDF-HSI (Figure 11, Appendix A). In the present study, cellular uptake experiments were not conducted for all the MPs investigated. The density of MPs may play a role in the effective exposure to cells. PETE, PMMA, and nylon MPs tested in this study have greater density than the density of water, whereas PS, PE, and PP MPs have lower density than the density of water [104]. Thus, compared with the other MPs tested, PETE, PMMA, and nylon have better chances to interact with cells in a submerged condition. However, this study did not investigate the specific impact of material density on cellular interaction and, consequently, on toxicity. The published studies also suggest that, in general, PETE is more harmful than PE, PP, or PS [105,106]. More studies are needed to fully appreciate the role of physical, chemical, and structural properties of MPs in toxicity, as other small-sized MPs investigated in the present study did not induce a similar toxicity response.

In the present study, MN formations were observed following MP exposure, with 100 nm PS showing the highest potency, followed by PETE and PMMA in the potency order of 100 nm PS < PETE < PMMA. MN formation has been documented following MP exposure in several in vitro studies. The mechanism by which MP exposure leads to MN formation is not well understood. Micro and nanoplastics may induce MN formation in different ways. It was shown that 3000 µm PS, PETE, and PE may induce MN formation in a potency order of PS < PETE < PE, which is in agreement with the findings of the current study [107]. Clastogenic events (chromosome breaks) or aneugenic events (chromosome loss) have been suggested to play a role in inducing MN formation. The MN formation after exposure to PE were not significant in this study, probably due to the lack of uniform size distribution. It has been known that with a non-homogenous particle size distribution, the smaller particles tend to agglomerate with the larger particles and thus the impact of smaller particles are not observed. It has been suggested that nanoplastics may interact with chromosomes before nuclear formation during cell division. Particle-induced genotoxicity may occur by direct particle–DNA interaction (primary mechanism) or the indirect interaction of particle–DNA through other molecules (secondary genotoxicity) [108]. Many nano and micro size particles have been shown to induce genotoxicity or ROS/oxidative stress and inflammation in vitro and in vivo [49,50,109,110].

The cellular uptake of spherical MPs ≤ 1 μm with smooth surfaces is expected to take place by endocytic or phagocytotic pathways, whereas MPs with irregular edges may be recruited by membrane piercing [82,111,112]. Membrane damage may result in reduced metabolic activity, organelle dysfunction, or increased DNA damage [113]. In a recent study, an aged 10 µm PS MP, possibly exhibiting irregular surfaces, was shown to induce higher DNA damage compared with the similar size pristine PS MPs of a uniform size [114]. In another study, PE MPs with irregular surface morphology were shown to induce higher cytotoxicity, inflammatory response, hemolysis, and ROS production at high concentrations compared with their counterparts with smooth surface morphologies [115]. Thus, several factors including size, shapes, chemical composition, and others may contribute to MP-induced toxicity. Moreover, the properties associated with the toxicity and the underlying mechanisms may differ depending on the type of endpoint tested and the toxicity models used.

## 5. Conclusions

Lab scale methods for generating MPs from regularly used and littered plastic items for toxicity testing were optimized in this study. A total of 11 unique MPs, both commercially purchased and lab-generated, exhibiting different sizes, shapes, and chemical compositions were tested for their potential to induce toxicity. The results showed that the lab-scale methods for MP generation are laborious. Although the methods produced a sufficient quantity of quality MPs for toxicity testing, the methods require further optimization to make them sustainable and applicable across laboratories. While each type of MP tested showed toxicity, i.e., reduction in cell viability, increased expression of cytokines, and increase in MN formation, of varying degrees, small-sized PETE MPs prepared from water bottles induced the maximum toxicity, showing positive responses in all endpoints assessed, suggesting small PETEs can potentially induce injury at the tissue level in vivo. Specifically, the ability of small PETE MPs to induce robust innate immune response, activating the interferon signaling pathways, suggests that plastic particles and fragments are perceived by cells by the similar mechanisms employed to recognize pathogens, leading to PAMP signaling. The non-specific interactions of MPs of heterogenous size and shapes with cells may be causing cell injury by activating the DAMP signaling and, consequently, triggering cell death, inflammatory cascade, and DNA damage, hallmark in vitro events indicative of potential in vivo tissue injury and tissue dysfunction. The present study was an attempt to understand if plastic materials in micron- and nano-size fractions are harmful. Further studies are required to systematically understand the plastics vs plastic chemical composition-induced toxicity and how size and shape play a role in such toxicity.

## Figures and Tables

**Figure 1 nanomaterials-14-01287-f001:**
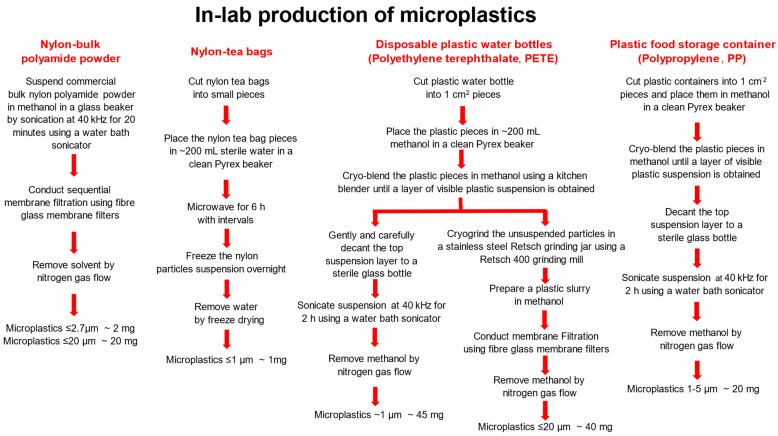
A schematic showing methods for in-lab generation of MPs.

**Figure 2 nanomaterials-14-01287-f002:**
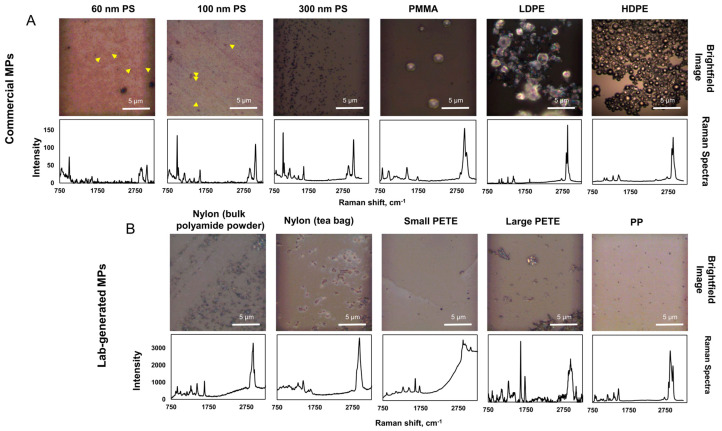
Raman characterization of commercial and in-lab generated MPs. The bright field images of MPs (**A**) and associated Raman spectrographs (**B**). The yellow arrows shows MPs in the nano size ranges.

**Figure 3 nanomaterials-14-01287-f003:**
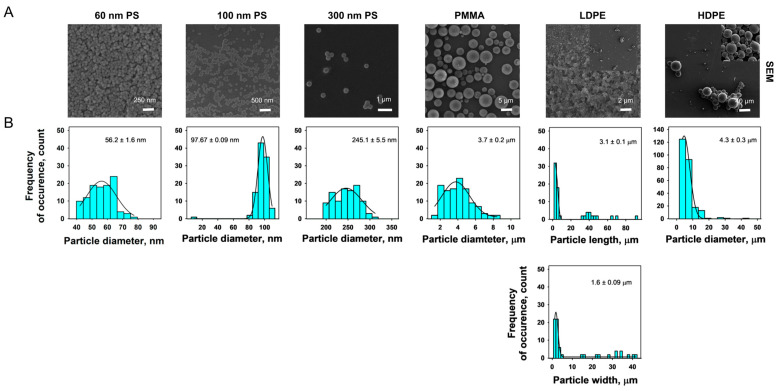
Representative scanning electron micrographs showing the morphology of commercial MPs used in this study (**A**) and size distribution histograms (**B**). The diameter of the spherical particles or primary particle lengths and widths for non-spherical particles were determined using ImageJ. The mean primary particle length and width were obtained from the histogram plots obtained from distribution of particle lengths and widths using SigmaPlot 15.

**Figure 4 nanomaterials-14-01287-f004:**
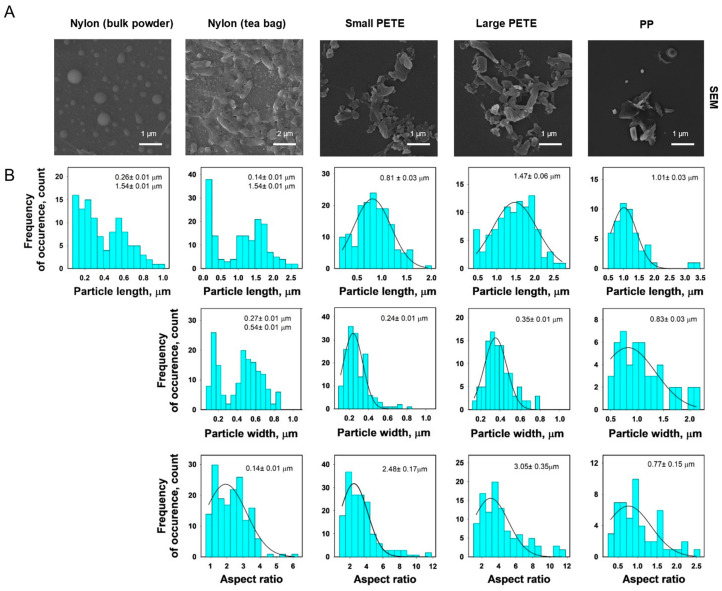
Representative scanning electron micrographs showing the morphology of the lab-generated MPs (**A**) and size distribution histogram plots (**B**). The diameter of the spherical particles or primary particle lengths and widths for non-spherical particles were determined using ImageJ. The mean primary particle length and width were obtained from the histogram plot obtained from distribution of particle lengths and widths using SigmaPlot 15.

**Figure 5 nanomaterials-14-01287-f005:**
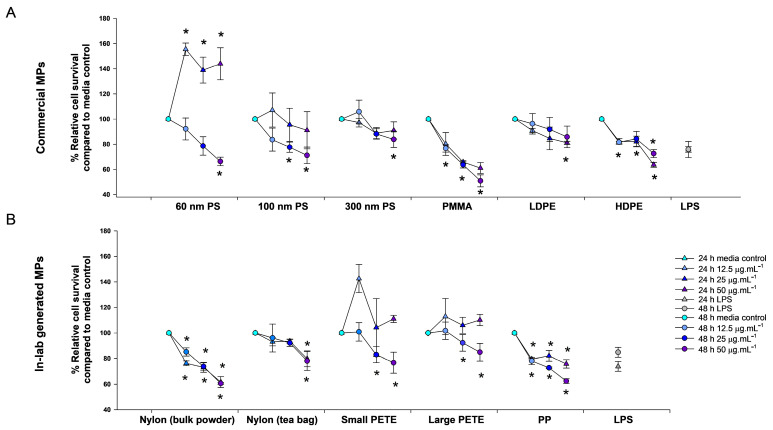
Relative cell survival post-24 h and 48 h exposure to 12.5, 25, and 50 µg mL^−1^ of commercial MPs or 1 µg mL^−1^ LPS compared with untreated controls (**A**) and post-exposure to 12.5, 25, and 50 µg·mL^−1^ of individual lab-generated MPs or 1 µg mL^−1^ LPS compared with untreated controls (**B**). The results are expressed as average % relative survival compared with the media controls and error bars depict standard errors. * Represents statistical significance. Statistical significance between the exposed samples and matched media control was determined (three biological replicates) by conducting a one-way ANOVA with Dunnett’s post hoc using *p* ≤ 0.05.

**Figure 6 nanomaterials-14-01287-f006:**
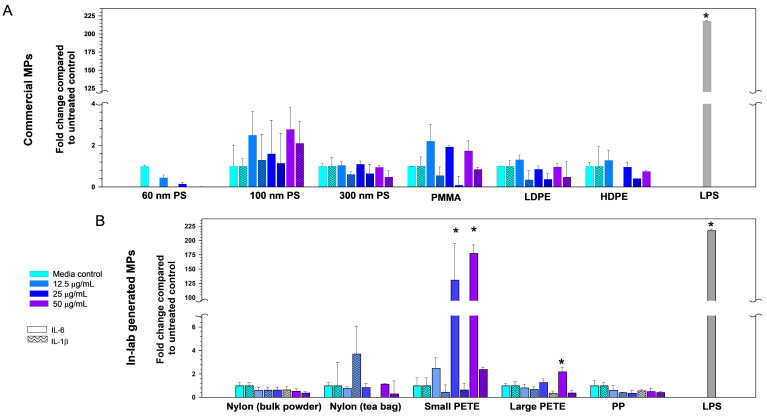
Analysis of pro-inflammatory proteins. IL-6 and IL-1 β expression in cell supernatant post-48 h exposure to 12.5, 25, and 50 µg·mL^−1^ of commercial MPs or 1 µg·mL^−1^ LPS via single cytokine ELISA (**A**) and IL-6 and IL-1 β expression levels in supernatant of cells post-48 h exposure to 12.5, 25, and 50 µg·mL^−1^ of lab-generated MPs or 1 µg mL−1 LPS via single cytokine ELISA (**B**). The results are expressed as average fold change compared with the media controls and error bars depict standard errors (three biological replicates, two technical replicates). * represents statistical significance. Statistical significance between the exposed samples and matched media control was determined by conducting a two-way ANOVA with a Dunnett’s post hoc using *p* ≤ 0.05.

**Figure 7 nanomaterials-14-01287-f007:**
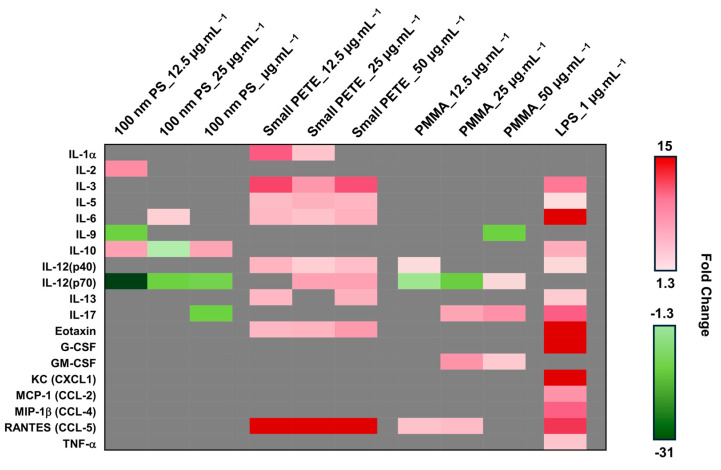
Analysis of pro-inflammatory cytokines via 23-plex cytokine ELISA—cytokine expression in supernatant of cells exposed to 12.5, 25, and 50 µg·mL^−1^ of 100 nm PS MP, PETE MP, and PMMA MP or 1 µg·mL^−1^ LPS for 48 h. Statistical significance (three biological replicates, two technical replicates) was calculated using Student’s *t*-test (*p* ≤ 0.1 and fold change compared with media control ≥ 1.3).

**Figure 8 nanomaterials-14-01287-f008:**
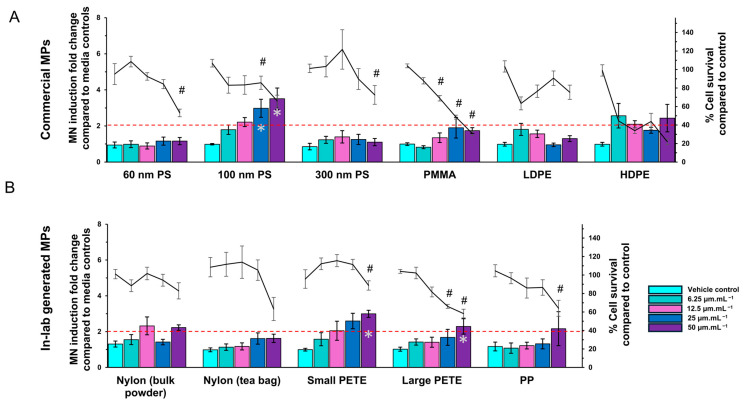
Micronuclei (MN) induction in treated versus untreated controls following 40 h post-exposure to commercial MPs (**A**) and lab-generated MPs (**B**). Bar graphs are the fold change in micronuclei induction compared with matched controls. The z-axis (line graph) presents the relative cell survival compared with untreated controls at 40 h post-exposure to commercial MPs (**A**) and lab-generated MPs (**B**). The error bars depict standard error. * represents statistical significance in MN induction and ^#^ represents statistical significance in % cell survival. Statistical significance between the exposed samples and matched media control was determined by conducting a one-way ANOVA with a Tukey’s post hoc multiple comparisons test using *p* ≤ 0.05. All experiments were conducted with three biological replicates and three technical replicates. The dashed red line represents the condition for genotoxicity (40% of the relative survival and 2-fold change threshold).

**Figure 9 nanomaterials-14-01287-f009:**
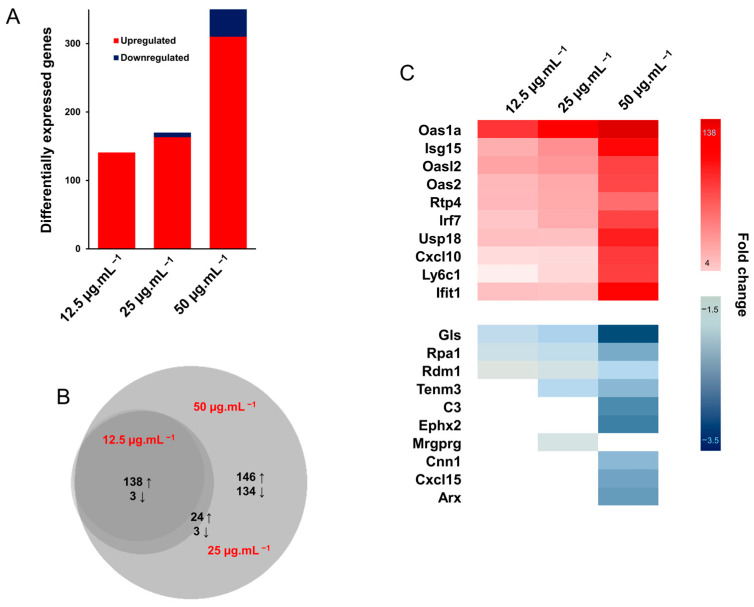
Differentially expressed genes post-small PETE exposure. The total numbers of DEGs following 48 h exposure to 12.5, 25, and 50 µg·mL^−1^ concentrations of PETE: (**A**) Red: upregulated and Blue: downregulated; Venn diagram showing the common DEGs between different concentration groups; (**B**) the upward and downward arrows depict up- and downregulated genes, and the heatmap showing the top 10 upregulated and downregulated DEGs (**C**).

**Figure 10 nanomaterials-14-01287-f010:**
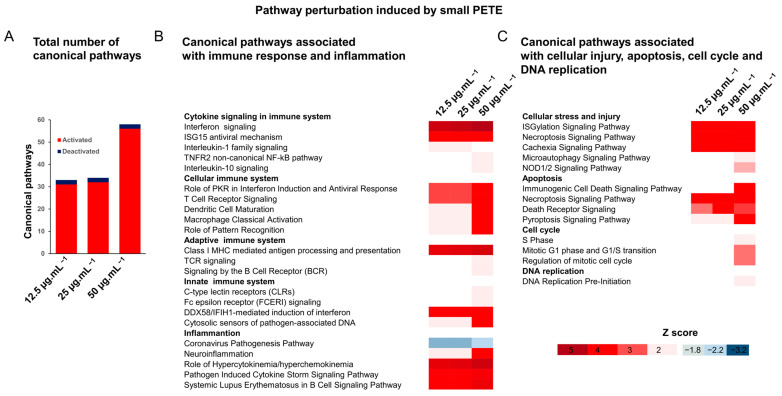
Pathway perturbation post-small PETE exposure. The total number of enriched canonical pathways (**A**); the enriched canonical pathways associated with immune and inflammation responses (**B**); and canonical pathways associated with cellular injury, apoptosis, cell cycle, and DNA replication (**C**).

**Figure 11 nanomaterials-14-01287-f011:**
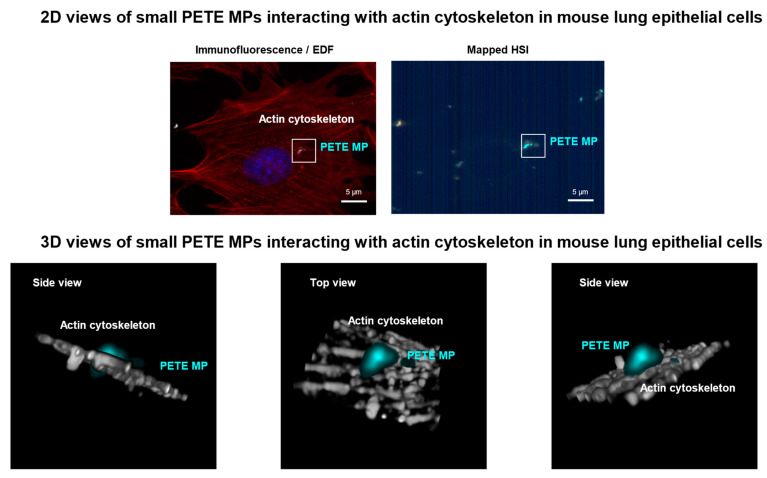
Two-dimensional view and three-dimensional rendering showing the small PETE interacting with the actin cytoskeleton in FE1-mouse lung epithelial cells. The 3D views represent the area marked by the white box in the 2D view. The 2D view (left) represents the of overlay of immunofluorescence image with EDF image in the same field of view (left) and corresponding hyperspectral images of the same field of view with mapped MPs (pseudo-colored in cyan).

**Table 1 nanomaterials-14-01287-t001:** DLS and ELS analysis results showing size distribution in water and in the cell culture medium.

MPs	Suspension Medium	Average Hydrodynamic Size, nm	PDI	Zeta Potential, mV
60 nm PS	MilliQ water	62.25 ± 0.36	0.0816	−39.16 ± 0.73
	Cell culture media	74.75 ± 0.89	0.0518	−15.72 ± 0.91
100 nm PS	MilliQ water	103.88 ± 0.77	0.0102	−46.88 ± 0.91
	Cell culture media	124.52 ± 0.68	0.0682	−14.38 ± 1.47
300 nm PS	MilliQ water	270.90 ± 1.72	0.0244	−50.38 ± 0.44
	Cell culture media	280.82 ± 2.16	0.0396	−14.66 ± 1.44
PMMA	MilliQ water	1810.33 ± 2079	0.66 ± 0.23	−49.62 ± 3.48
	Cell culture media	1092.98 ± 737	0.78 ± 0.15	−8.47 ± 0.36
LDPE	MilliQ water	315.1 ± 121	0.48 ± 0.07	−39.4 ± 4.14
	Cell culture media	342.34 ± 82.5	0.51 ± 0.09	−12.98 ± 0.68
HDPE	MilliQ water	3649.25 ± 1831	0.64 ± 0.25	−47.9 ± 4.23
	Cell culture media	1664.5 ± 300	0.48 ± 0.38	−5.03 ± 0.45
Nylon from bulk polyamide powder	MilliQ water	449.75 ± 11.26	0.44 ± 0.09	−31.18 ± 1.38
	Cell culture media	466.84 ± 15.13	0.49 ± 0.03	−11.46 ± 1.15
Nylon from tea bags	MilliQ water	698.62 ± 82.53	0.59 ± 0.06	−32.06 ± 0.68
	Cell culture media	223.58 ± 63.2	0.78 ± 0.12	−7.08 ± 0.73
Small PETE	MilliQ water	406 ± 5.06	0.28 ± 0.03	−39.3 ± 1.4
	Cell culture media	713.33 ± 27	0.53 ± 0.04	−17.4 ± 1.07
Large PETE	MilliQ water	1117.3 ± 383	0.61 ± 0.27	−41.98 ± 3.49
	Cell culture media	731.9 ± 107	0.57 ± 0.06	−15.46 ± 0.47
PP	MilliQ water	185.84 ± 44.86	0.40 ± 0.08	−52.28 ± 11.9
	Cell culture media	1809.04 ± 886.34	0.92 ± 0.1	−14.94 ± 0.9

## Data Availability

The data presented in this study are available on request from the corresponding author.

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
