# Peer review of "Polyethylene Terephthalate Microplastics Generated from Disposable Water Bottles Induce Interferon Signaling Pathways in Mouse Lung Epithelial Cells"

_nanomaterials, 2024, doi:10.3390/nano14151287_

Round 1

Reviewer 1 Report

Comments and Suggestions for Authors

The work of Luna Rahman, Andrew Williams, Dongmei Wu and Sabina Halappanavar named “Polyethylene terephthalate microplastics generated from disposable water bottles induce interferon signaling pathway in mouse lung epithelial cells” is devoted to assessing the consequences of the action one from most not unreasonably popular types of environmental pollution, i.e. microplastic pollution .

In this area, many studies have been carried out on the cytotoxicity and pro-inflammatory effects of microplastic particles on animal and human cells, as well as in vivo. However, this study is surprising in its complexity: combining the approaches of immunology, genetic toxicology, classical cytology, new methods of microscopy. The introduction is written in detail and clearly. There is a lot of quantitative data that clearly illustrates the scale of the problem.

The advantage of the work is not only a wide panel of tested parameters, but also the authors’ attempt to determine which intracellular pathways are activated or inhibited as a result of the action of microplastic particles (in particular, the signaling pathways of the innate and adaptive immune response, cellular stress and injury, apoptosis etc.). The results are presented in easy-to-read figures and are described and discussed in detail in the text. The infiltration of microplastic particles into the cytoskeleton has been shown. An interesting applied result is that in the cultivation medium the module zeta potential is much lower than in MiliQ water. The dependence of cytotoxicity has been characterized microplastics from the composition of particles. PETE particles is most cytotoxic. Proinflammatory activity decreases in the order “PETE>100 nm PS>PMMA> large PETE”.

I believe that the work can be published after a number of additions.

 In the introduction, it is advisable to add examples of the adverse effects of microplastics on the human cardiovascular system (for example, the negative effects of MPs are shown in 10.1056/NEJMoa2309822 and 10.3390/nano11020496).

 line 234 It is advisable to indicate the method of sample preparation before SEM.

 line 238 The version of Image J should be indicated. If a special Marcos or plugin was used, it is also advisable to indicate this.

 line 247 The concentration of L-glutamine is not specified. If glutamine has already been added to the medium, it is advisable to indicate this with an indication of the concentration. If L-glutamine was added separately, the concentration and manufacturer should be indicated.

 line 250 Different fonts. Please, check fonts in the all text.

 line 255 Please indicate the power and wavelength of ultrasonic treatment, if possible.

 line 264 For lipopolysaccharide (LPS), the species, strain and serotype (if any) of the producing bacterium must be indicated. If it is a commercially available LPS, the manufacturer must be identified. If it was obtained independently, it is necessary to describe the methods for obtaining and analyzing the purity. The immunogenicity of LPS varies significantly even between different strains of the same species, so this information is critical.

 line 269. Manufacturers are not listed for all reagents, instruments and software, for example, Trypan blue. Please add the missing information.

 line 270. Please indicate the concentration of trypsin used to separate the cells and whether EDTA was added (and at what concentration).

 line 336 Methyl methanesulfonate concentration should be indicated.

 line 390 The space was missed. Please check in all text.

 lines 399 and 408 Different spellings “p values” and “p - value”. To accept the alternative hypothesis, the strict inequality p<0.05 must be observed. Please check and revise.

 Authors should indicate under what conditions Student's t-test was chosen and under what ANOMA. How was the normality of the distribution checked?

 Figures 3B and 4B. Please add the units on the ordinate axis.

 Figure 6. It is advisable to indicate the initial concentrations of cytokines (in pg/ml ). This data may be included in applications.

 Figure 11. Scale bars need to add to the pictures.

 It is advisable to add a decoding for the abbreviation PAMP.

 In the discussion, the authors would like to add potential strategies based on their findings to protect against the negative effects of microplastics. For example, how effective will the use of antioxidants described in the literature be (10.3389/fenvs.2021.811466).

 Best regards

Author Response

Response to comments by Reviewer 1_round 1

Thank you for your kind consideration and valuable comments. The comments have been addressed in detail in a point by point manner.

Comment 1

In the introduction, it is advisable to add examples of the adverse effects of microplastics on the human cardiovascular system (for example, the negative effects of MPs are shown in 10.1056/NEJMoa2309822 and 10.3390/nano11020496).

Response 1:

Line 84 has been revised as follows to include the adverse effects of microplastics in humans, including the cardiovascular system.

“MPs from the target lung tissue may translocate to other organs. In a recent clinical study, higher rate of infarction, nonfatal stroke, or death was observed among asymptomatic patients with internal carotid artery stenosis detected with MPs in their carotid artery plaques compared to the patients not detected with MPs in their carotid artery plaques [31]. In the past, the outbreak of Interstitial lung disease among textile workers at nylon flock industries situated in Ontario, Canada and in Rhode Island, United States of America during 1992-1996 was associated with exposure to respirable size fragmented nylon [32, 33]. These results suggest a potential for tissue exposure to MPs and adverse effects in humans.”

References

  1. Marfella R, Prattichizzo F, Sardu C, Fulgenzi G, Graciotti L, Spadoni T, et al. Microplastics and Nanoplastics in Atheromas and Cardiovascular Events. N Engl J Med. 2024;390(10):900-10; doi: 10.1056/NEJMoa2309822.
  2. Lougheed MD, Roos JO, Waddell WR, Munt PW. Desquamative interstitial pneumonitis and diffuse alveolar damage in textile workers: potential role of mycotoxins. Chest. 1995;108(5):1196-200.
  3. Flock Worker's Lung: Chronic Interstitial Lung Disease in the Nylon Flocking Industry. Annals of Internal Medicine. 1998;129(4):261-72; doi: 10.7326/0003-4819-129-4-199808150-00001 %m 9729178.

Comment 2

line 234 It is advisable to indicate the method of sample preparation before SEM.

Response 2:

Line 255: The text is revised as follows to describe the sample preparation method for SEM analysis.

“Suspensions of MPs at a concentration of 50 µg.mL-1 were prepared in ethanol in clean glass bottles by sonicating for 5 minutes using a Bransonic digital ultrasonic water bath at 40 kHz at room temperature (RT). About 2-3 µL of each MP suspension were deposited on an aluminum stub and dried for 20 minutes at RT. Since the MPs were not conducting materials, dried MPs on stubs were sputter coated with gold film to a thickness of 10 nm low vacuum sputter coater (Quorum, Q150T ES, Ashford, Kent, UK).”

Comment 3

line 238 The version of Image J should be indicated. If a special Marcos or plugin was used, it is also advisable to indicate this.

Response 3:

The version of Image J used has been added in line 262.

“The SEM images of at least 100 individual objects (particles, fibres, fragments) were used to determine the diameter, length and width of the commercially purchased and lab-generated MPs using Image J (version 1.53g) software [62].”

Comment 4

line 247 The concentration of  L-glutamine is not specified. If glutamine has already been added to the medium, it is advisable to indicate this with an indication of the concentration. If L-glutamine was added separately, the concentration and manufacturer should be indicated.

Response 4:

Line 271:  The text is modified as follows with L-Glutamine concentrations indicated:

“Cells were cultured in Dulbecco’s Modified Eagle’s Medium Nutrient Mixture (DMEM) made up of F12 HAM (1:1) culture media with 365 mg.L-1 L-Glutamine, (Life Technologies, Burlington, Ontario, Canada) supplemented with 2% fetal bovine serum (FBS, Life Technologies, Burlington, Ontario, Canada), 1 ng.mL-1 human epidermal growth factor (EGF, Life Technologies, Burlington, Ontario, Canada), 100 U.mL-1 penicillin G, and 100 µg.mL-1 streptomycin (Life Technologies, Burlington, Ontario, Canada).”

Comment 5

line 250 Different fonts. Please, check fonts in the all text.

Response 5:

Line 277:  The font style and the size are now consistent.

Comment 6

line 255 Please indicate the power and wavelength of ultrasonic treatment, if possible.

Response 6:

The specific power has been added; however, there is no specific wavelength to indicate. 

Line 283: “Sonicated and suspended MPs in MilliQ water or in cell culture media using a Bransonic digital ultrasonic water bath at 40 kHz were diluted to a final concentration of 50 µg.mL-1, 100 µg.mL-1 or 1 mg.mL-1 (Table S1 details the sample preparation) and an aliquot of the suspensions was used for measuring the hydrodynamic diameter, polydispersity index and the zeta potential.”

Comment 7

line 264 For lipopolysaccharide (LPS), the species, strain and serotype (if any) of the producing bacterium must be indicated. If it is a commercially available LPS, the manufacturer must be identified. If it was obtained independently, it is necessary to describe the methods for obtaining and analyzing the purity. The immunogenicity of LPS varies significantly even between different strains of the same species, so this information is critical.

Response 7:

The requested information is added as follows:

Line 290: “After 24 h, cells were treated with media only or media containing MPs at concentrations ranging from 6.25 µg.mL-1 to 50 µg.mL-1 or with 1 µg.mL-1 of lipopolysaccharide (LPS, Escherichia coli 055: B5, Sigma-Aldrich, Oakville, Ontario, Canada) for 24 h and 48 h.”

Comment 8

line 269. Manufacturers are not listed for all reagents, instruments and software, for example, Trypan blue. Please add the missing information.

Response 8:

Line 300: The text has been modified as follows:

“Suspended cells (10 µL) were combined with an equal volume of Trypan blue stain (Life Technologies, Burlington, Ontario, Canada) and incubated for 5-6 minutes at RT. “

For all other reagents, manufacturers are listed.

Comment 9

line 270. Please indicate the concentration of trypsin used to separate the cells and whether EDTA was added (and at what concentration).

 Response 9:

Line 297: The text is modified as follows:

“In brief, cells were trypsinized post-exposure by incubating with 150 µL of 0.25% Trypsin-EDTA (disodium ethylenediaminetetraacetic acid) and suspended in cell culture media.”

Comment 10

line 336 Methyl methanesulfonate concentration should be indicated.

 Response 10:

Line 370: The text is modified as follows:

“Methyl methanesulfonate (MMS) (Sigma–Aldrich, Oakville, Ontario, Canada), a known genotoxicant at a concentration of 500 µM was used as a positive control.”

Comment 11

 line 390 The space was missed. Please check in all text.

Response 11:

The text has been checked for editorial corrections.

 Comment 12

lines 399 and 408 Different spellings “p values” and “- value”. To accept the alternative hypothesis, the strict inequality p<0.05 must be observed. Please check and revise.

Response 12:

Line 433: The text is corrected as follows:

“For statistical significance, p-values were estimated using permutation methods, which were adjusted for multiple comparison with false discovery rate test corrections [75].”

Comment 13

Authors should indicate under what conditions Student's t-test was chosen and under what ANOMA. How was the normality of the distribution checked?

Response 13:

To address the reviewers’ comments, the statistical significance of cell viability results was redetermined using a Kruskal-Wallis One Way Analysis of Variance (ANOVA) on Ranks to determine the significance between the exposed samples and matched media controls. When the condition for normality test (Shapiro-Wilk test) and equal variance passed, a One Way ANOVA with Dunnett’s post hoc was carried out.

 Th text in the manuscript has been revised to reflect the changes.

Line 308: “A Kruskal-Wallis One Way Analysis of Variance (ANOVA) on Ranks was conducted using SigmaPlot 15 to determine the significance between the exposed samples and matched media control. If the conditions for normality test (Shapiro-Wilk test) and equal variance passed, a One Way ANOVA with Dunnett’s post hoc was carried out to determine the significance between the exposed samples and matched media controls (three biological replicates) using p ≤ 0.05.”

Figure 5 and the associated legend is revised as follows:

“Figure 5. Relative cell survival post-24 h and 48 h exposure to 12.5, 25, and 50 µg mL−1 of commercial MPs or 1 µg mL−1 LPS compared to untreated controls (A) and post-exposure to 12.5, 25, and 50 µg.mL−1 of individual lab-generated MPs or 1 µg mL−1 LPS compared to untreated controls (B). The results are expressed as average % relative survival compared to the media controls and error bars depict standard errors. * Represents statistical significance. Statistical significance between the exposed samples and matched media control was determined (three biological replicates) by conducting a One Way ANOVA with Dunnett’s post hoc using p ≤ 0.05.”

Accordingly, the results sections are revised as shown below:

Line 617:

“After 48 h post-exposure to 60 nm PS, at a concentration of 50 µg.mL-1, the % relative survival displayed statistically significant decrease, measuring 66.3%. However, the decreases were not statistically significant at the lower concentrations of 12.5 µg.mL-1 and 25 µg.mL-1, where the % relative survivals remained relatively high at 78.6% and 92.2% respectively.”

Line 644:

  “The decrease in relative survival was statistically significant for all concentrations of PMMA at 48 h post-exposure, for high concentration groups of LDPE and for all concentrations of HDPE for both time points.”

Line 654:

“The results were significant for all concentrations for the nylon MPs prepared from bulk powder and at high concentrations of nylon MPs from tea bags.”

Comment 14

Figures 3B and 4B. Please add the units on the ordinate axis.

 Response 14:

Y-Axis title for the 3B and 4B has been changed from “Frequency” to “Frequency of occurrence, count”.

Comment 15

Figure 6. It is advisable to indicate the initial concentrations of cytokines (in pg/ml ). This data may be included in applications.

Response 15:

The cytokine concentration in the supernatants of FE1 cells exposed to commercial and in-lab generated MPs are presented in the Figure below.

Figure S3. Analysis of pro-inflammatory proteins. IL-6 and IL-1 β expression in cell supernatant post-48 h exposure to 12.5, 25, and 50 µg.mL−1 of commercial MPs via single cytokine ELISA (A) and IL-6 and IL-1 β expression levels in supernatant of cells post-48 h exposure to 12.5, 25, and 50 µg.mL−1 of lab-generated MPs via single cytokine ELISA (B).  * represents statistical significance. Statistical significance between the exposed samples and matched media control (three biological replicates, two technical replicates) was determined by conducting a two-way ANOVA with a Dunnett’s post hoc using p ≤ 0.05.

We prefer to express the results in fold-change, as that is how all our previous results were presented in peer-reviewed publications. However, we have made a separate Figure S3 (new) to indicate the specific concentrations of cytokines.

A sentence has been added at line 684.

“The concentration of cytokines in the supernatants following MP exposure are presented in Figure S3.”

Comment 16

Figure 11. Scale bars need to add to the pictures.

Response 16:

 Scale bars have been added to 2D images. It is not possible to add scale bars in the 3D images.

Comment 17

It is advisable to add a decoding for the abbreviation PAMP.

Response 17 :

The abbreviation PAMP was decoded in line 923.

“However, in the present study, the lab-generated MPs did not show endotoxin contamination, suggesting that cells may be perceiving MPs as PAMPs (Pathogen Associated Molecular Patterns).”

Comment 18

In the discussion, the authors would like to add potential strategies based on their findings to protect against the negative effects of microplastics. For example, how effective will the use of antioxidants described in the literature be (10.3389/fenvs.2021.811466).

 Best regards

Response 18 :

While the use of antioxidants may improve the reproductive health of aquatic animals which is shown to be impacted by MPs [1], the best way to control the negative effect of MPs on human health and environment is to reduce the use and production of plastic materials to essential reasons only [2], especially the single use plastic products or use alternate biodegradable plastics [3].

References

  1. El-Din H. Sayed A, Hamed M, Ismail RF. Natural Antioxidants can Improve Microplastics-Induced Male Reproductive Impairment in the African Catfish (Clarias Gariepinus). Frontiers in Environmental Science. 2022;9; doi: 10.3389/fenvs.2021.811466.
  2. Peng X, Jiang Y, Chen Z, Osman AI, Farghali M, Rooney DW, Yap P-S. Recycling municipal, agricultural and industrial waste into energy, fertilizers, food and construction materials, and economic feasibility: a review. Environmental Chemistry Letters. 2023;21(2):765-801.
  3. Farghali M, Mohamed IMA, Osman AI, Rooney DW. Seaweed for climate mitigation, wastewater treatment, bioenergy, bioplastic, biochar, food, pharmaceuticals, and cosmetics: a review. Environ Chem Lett. 2023;21(1):97-152; doi: 10.1007/s10311-022-01520-y.

The research on MP induced human effects is in its early days and not all adverse impacts of MPs exposure have been conclusively shown. Therefore, it may be premature to discuss the protective strategies targeting the mechanisms that are yet to be fully discovered.

Reviewer 2 Report

Comments and Suggestions for Authors

Manuscript was prepared well with several characterization techniques and excellent figures quality. Manuscript will be accepting after minimal corrections

- Line 77 change “97% and 49%” to “97 and 49%”

- Introduction has a lot of information (more than two pages) try to summarize

- Lines 161, 168, and 169 change “60 nm, 100 nm, and 300 nm” to “60, 100, and 300 nm” and the same recommendation for all manuscript

- 2. Materials and Methods. Include information about purification materials or sentence “used s received”

- Line 583, why selected “After 48 h”

- Line 589 change “83.6%, 77.6% and 71.1%” to “83.6, 77.6, and 71.1%” and the same recommendation for all manuscript (lines 606-611, etc.)

- Include important results in the conclusion part

Author Response

Response to comments by Reviewer 2_round 1

Thank you for your kind consideration and valuable comments. The comments have been addressed in detail in a point by point manner.

Comment 1

- Line 77 change “97% and 49%” to “97 and 49%”

Response 1:

It may be best to repeat the units for clarity.

Comment 2

- Introduction has a lot of information (more than two pages) try to summarize

Response 2:

Given the newness of the research area and scattered research results, the authors are of the opinion that the length of the introduction section is appropriate and maybe needed.

Comment 3

- Lines 161, 168, and 169 change “60 nm, 100 nm, and 300 nm” to “60, 100, and 300 nm” and the same recommendation for all manuscript

Response 3:

It may be best to repeat the units for clarity. However, in this case, we have made the requested revision.

Line 169:

Aqueous suspension of unmodified PS beads of 60, 100, and 300 nm (National Institute of Standards and Technology (NIST), Maryland, United States); Low-density polyethylene (LDPE) large beads (1000-5000 nm, non-nano certified reference material, Nanochemazone™, Alberta, Canada), High-density polyethylene (HDPE) large beads (1000-5000 nm, non-nano certified reference material, Nanochemazone™, Alberta, Canada), bulk polyamide nylon powder (5-50 µm, non-nano certified reference material, Sigma Aldrich, Mississauga, Ontario, Canada) and polymethyl methacrylate (PMMA, 15,00-11,500 nm , Cospheric, California, USA), were purchased.

Line 176:

The 60, 100, and 300 nm PS suspensions contained an emulsifier (possibly, sodium lauryl sulfate or sodium 1-dodecanesulfonate and 1-dodecanol at a concentration of < 0.05%) as well as an electrolyte (∼0.02%) to prevent agglomeration.

Line 180:

The 100 and 300 nm PS suspensions contained 50 ppm sodium azide to retard the growth of algae and bacteria.

Comment 4

- 2. Materials and Methods. Include information about purification materials or sentence “used s received”

Response 4:

Materials were used as received for the current study. A sentence has been added at line 181,

“All commercially purchased materials were used as received without further purification.”

Comment 5

- Line 583, why selected “After 48 h”

Response 5:

Line 896:

“At 48 h post-exposure, most MPs induced a response in the context of various endpoints tested and thus, 48 h timepoint was chosen for the gene expression analysis.” 

Comment 6

- Line 589 change “83.6%, 77.6% and 71.1%” to “83.6, 77.6, and 71.1%” and the same recommendation for all manuscript (lines 606-611, etc.)

Response 6:

We disagree with the suggested format changing. The units must repeated for clarity. Please see the response for your comment 1.

Comment 7

- Include important results in the conclusion part

Response 7:

The authors believe that all results have been accurately captured. However, to address the reviewer’s comment, a sentence has been added (Line 1026) to the conclusion section.

Lab scale methods for generating MPs from regularly used and littered plastic items for toxicity testing were optimized in this study. A total of 11 individual MPs, both commercially purchased and lab-generated, exhibiting different sizes, shapes and chemical composition were tested for their potential to induce toxicity. The results showed the lab-scale methods for MP generation are laborious. Although the methods produced sufficient quantity of quality MPs for toxicity testing, the methods require further optimization to make them sustainable and applicable across laboratories. While individual MPs tested showed toxicity, i.e., reduction in cell viability, increased expression of cytokines and increase in MN formation, of varying degree, small sized PETE MPs prepared from water bottles induced the maximum toxicity, showing positive response in all endpoints assessed, suggesting small PETEs can potentially induce injury at the tissue level in vivo. Specifically, the ability of small PETE MPs to induce robust innate immune response, activating the interferon signaling pathways, suggests that plastic particles and fragments are perceived by cells by the similar mechanisms employed to recognize pathogens, leading to PAMP signaling. The non-specific interactions of MPs of heterogenous size and shapes with cells may be causing cell injury activating the DAMP signaling and consequently, triggering the cell death, inflammatory cascade and DNA damage, hallmark in vitro events indicative of potential in vivo tissue injury and tissue dysfunction. The present study was an attempt to understand if plastic materials in micron and nano size fraction are harmful. Further studies are required to systematically understand the plastics vs plastic chemical composition induced toxicity and how size and shapes play a role in such toxicity.

Reviewer 3 Report

Comments and Suggestions for Authors

The authors present a comprehensive study with astonishingly high amount of data like production of the microparticles and directly application of these in toxic assays followed by micronuclei test and gene expression and signaling investigation. The study is very well done and very well explained. The results are more significant and worrying than I expected. I think that the work should be published after just a minor revision taking the points below into account.

11.      Did the authors use the right symbol in page 2 line 41? 5mm would be milli- and not microplastic

22.      Page 2 line 48, for this claim I expect 3 references

33.      particles.m-2 should it be not particles /m2??

44.      particles.m-3 here particles /m³?

55.      Page 2 line 86 reference missing

66.      Page 3 line 107 reference missing

77.      Page 3 line 126 reference missing

88.      Power of the ultrasound bath is missing

99.      Why are some methods utilizing blending, some microwaving and more? The authors should mention why the choices are as they are? Were pre-tests done?

110.   Why were endotoxin levels measured? The explanation is missing in methods and materials section.

111.   ImageJ requires as part of the licence the citation of their publication.1 Furthermore, the authors should state the software version they utilized.

112.   Page 10 line 430 2 should be subscript

113.   The authors utilize two different equation / unit expression styles, please unify

114.   Figure 3: The reason for an additional width for the LDPE should be mentioned in the Figure caption.

115.   Figure 11 scale bar missing

116.   “. (reviewed in Alijagic et. al.)[78].”

117.   The authors should also mention, that many novel drug delivery systems will utilize negatively charged polymer systems2 and that the particles they investigate might be also get into the body by drinking, or by injecting (particles are also within medical devices like needles and breathing systems). Exposure from these systems also can get into contact with lung cells.

References

(1)        Abràmoff, M. D.; Magalhães, P. J.; Ram, S. J. Image Processing with ImageJ. Biophotonics Int. 2005, 11 (7), 36–43.

(2)        Hu, N.; Frueh, J.; Zheng, C.; Zhang, B.; He, Q. Photo-Crosslinked Natural Polyelectrolyte Multilayer Capsules for Drug Delivery. Colloid. Surf., A 2015, 482, 315–323. https://doi.org/10.1016/j.colsurfa.2015.06.014.

Author Response

Response to comments by Reviewer 3_round 1

Thank you for your kind consideration and valuable comments. The comments have been addressed in detail in a point by point manner.

Comment 1

Did the authors use the right symbol in page 2 line 41? 5mm would be milli- and not microplastic

Response 1:

According to the traditional definition of microplastics, size 5 mm or smaller is correct.

Comment 2

  1. Page 2 line 48, for this claim I expect 3 references

Response 2:

Line 46: References are inserted.

However, studying airborne MP pollution and its potential inhalation toxicity in humans poses significant challenges [5-7].

References

  1. Cao G, Cai Z. Getting Health Hazards of Inhaled Nano/Microplastics into Focus: Expectations and Challenges. Environmental Science & Technology. 2023;57(9):3461-3; doi: 10.1021/acs.est.3c00029.
  2. Kelly FJ, Fussell JC. Toxicity of airborne particles—established evidence, knowledge gaps and emerging areas of importance. Philosophical Transactions of the Royal Society A: Mathematical, Physical and Engineering Sciences. 2020;378(2183):20190322; doi: 10.1098/rsta.2019.0322.
  3. Alqahtani S, Alqahtani S, Saquib Q, Mohiddin F. Toxicological impact of microplastics and nanoplastics on humans: understanding the mechanistic aspect of the interaction. Front Toxicol. 2023;5:1193386; doi: 10.3389/ftox.2023.1193386.

Comment 3

  1. particles.m-2 should it be not particles /m2??

Response 3:

For consistency, it is written as particles.m-2..

Comment 4

  1. particles.m-3 here particles /m³?

Response 4:

The style is maintained for consistency.

Comment 5

  1. Page 2 line 86 reference missing

Response 5:

Line 94: References are inserted.

“PS MPs smaller than 100 µm have been found accumulated in larvae and guts of fish and have been shown to induce oxidative stress, inflammation and thereby negatively impacting liver metabolism, the reproductive system and neuronal development in fish [34-36].”

References

  1. Wan Z, Wang C, Zhou J, Shen M, Wang X, Fu Z, Jin Y. Effects of polystyrene microplastics on the composition of the microbiome and metabolism in larval zebrafish. Chemosphere. 2019;217:646-58; doi: https://doi.org/10.1016/j.chemosphere.2018.11.070.
  2. Chen L, Hu C, Lai NL-S, Zhang W, Hua J, Lam PK, et al. Acute exposure to PBDEs at an environmentally realistic concentration causes abrupt changes in the gut microbiota and host health of zebrafish. Environmental Pollution. 2018;240:17-26; doi: 10.1016/j.envpol.2018.04.062.
  3. Qiao R, Deng Y, Zhang S, Wolosker MB, Zhu Q, Ren H, Zhang Y. Accumulation of different shapes of microplastics initiates intestinal injury and gut microbiota dysbiosis in the gut of zebrafish. Chemosphere. 2019;236:124334; doi: 10.1016/j.chemosphere.2019.07.065.

Comment 6

  1. Page 3 line 107 reference missing

Response 6:

Line 112: References have been cited,

“Inflammation and perturbations in metabolic homeostasis have been observed in high-fat diet obese mice [44] or in ICR mice [45] following six-weeks oral exposure to 1-5 μm PS MPs, while cardiac fibrosis was reported in male Wistar rats following 90-days oral exposure to spherical 0.5 μm PS MPs [46].”

References

  1. Lee AG, Kang S, Yoon HJ, Im S, Oh SJ, Pak YK. Polystyrene Microplastics Exacerbate Systemic Inflammation in High-Fat Diet-Induced Obesity. International Journal of Molecular Sciences. 2023. doi: 10.3390/ijms241512421.
  2. Lu L, Wan Z, Luo T, Fu Z, Jin Y. Polystyrene microplastics induce gut microbiota dysbiosis and hepatic lipid metabolism disorder in mice. Science of The Total Environment. 2018;631-632:449-58; doi: https://doi.org/10.1016/j.scitotenv.2018.03.051.
  3. Li Z, Zhu S, Liu Q, Wei J, Jin Y, Wang X, Zhang L. Polystyrene microplastics cause cardiac fibrosis by activating Wnt/β-catenin signaling pathway and promoting cardiomyocyte apoptosis in rats. Environmental Pollution. 2020;265:115025; doi: 10.1016/j.envpol.2020.115025.

Comment 7

  1. Page 3 line 126 reference missing

A reference has been cited in line 134,

“Moreover, there is a scarcity of research on the toxicity of other common plastics such as PP, PE, and polyamides (PA) [57].”

Reference

  1. 57. Kokalj AJ, Hartmann NB, Drobne D, Potthoff A, Kühnel D. Quality of nanoplastics and microplastics ecotoxicity studies: Refining quality criteria for nanomaterial studies. Journal of Hazardous Materials. 2021;415:125751; doi: https://doi.org/10.1016/j.jhazmat.2021.125751.

Response 7:

Comment 8

  1. Power of the ultrasound bath is missing

Response 8:

The power of ultrasound bath is added in line 282.

Line 282: Sonicated and suspended MPs in MilliQ water or in cell culture media using a Bransonic digital ultrasonic water bath at 40 kHz were diluted to a final concentration of 50 µg.mL-1, 100 µg.mL-1 or 1 mg.mL-1 (Table S1 details the sample preparation) and an aliquot of the suspensions was used for measuring the hydrodynamic diameter, polydispersity index and the zeta potential.

In addition, the power of ultrasound bath was already mentioned in the following lines.

Line 189: About 200 mg of the bulk nylon polyamide powder was sonicated in 1 L Methanol (Omnisolve, 99.9%, Sigma Aldrich, Oakville, Ontario) in a clean autoclaved glass bottle for 20 minutes using a Bransonic digital ultrasonic water bath (model 2800, Branson, Danbury, USA) at 40 kHz at room temperature.

Line 208: Using a kitchen steel hand blender, the PETE and PP plastic pieces in methanol were blended separately until a layer of visible plastic suspension was obtained and further sonicated in an ice bath at 40 kHz for 2 h, using a Bransonic digital ultrasonic water bath sonicator (model 2800, Branson, Danbury, USA) to break up the aggregates until a uniform suspension of MPs was evident.

Comment 9

  1. Why are some methods utilizing blending, some microwaving and more? The authors should mention why the choices are as they are? Were pre-tests done?

Response 9:

The authors have added a few sentences starting at line 219 in the method section to clarify the reason behind choosing different techniques to generate different types of microplastics.

“The choice of methods for degradation of plastics was based on different conditions under which these plastic types may become brittle and break, keeping in mind the actual degradation processes in the environment. Plastics such as PETE and PP become brittle when stored at very cold temperature, which could be further subjected to cryoblending for generating MPs. However, nylon from teabags is resistant to cold temperatures and can withstand high heat, leading to the use of microwave oven for breaking nylon tea bags. The processes employed also reflected the ease with which plastics could be broken down to generate MPs.”

Comment 10

  1. Why were endotoxin levels measured? The explanation is missing in methods and materials section.

Response 10:

A sentence has been added at line 228,

“Although the plastic items chosen for generating MPs in the laboratory were thoroughly cleaned, they were previously used and exposed to household environmental conditions. Consequently, the lab-generated MPs underwent an endotoxin test, as endotoxins are abundant in the environment and can activate immune responses in cells, potentially obscuring the true effects of exposure to the MPs.”

Comment 11

  1. ImageJ requires as part of the license the citation of their publication.1Furthermore, the authors should state the software version they utilized.

Response 11:

The version of Image J has been included in the sentence starting at line 262.

“The SEM images of at least 100 individual objects (particles, fibres, fragments) were used to determine the diameter, length and width of the commercially purchased and lab-generated MPs using Image J (version 1.53g) software [62].”

Reference

  1. Abràmoff MD, Magalhães PJ, Ram SJ. Image processing with ImageJ. Biophotonics international. 2004;11(7):36-42.

Comment 12

  1. Page 10 line 430 2 should be subscript,

Response 12:

Line 463 has been corrected.

“FE1 cells were seeded on coverslips at a concentration of 60,000 cells per well in a 6-well plate and incubated at 37°C with 5% CO2 for 24 h to settle.”

Comment 13

  1. The authors utilize two different equation / unit expression styles, please unify

Response 13:

Unit expression styles have been checked.

Line 304 has been corrected as,

“Percent relative cell survival compared to media treated time-matched controls were determined from the ratio of the number of live cells in a sample and the average number of live cells in the media controls (the percentage of relative survival (% relative survival = [Number of live cells.cm-2 of samples]/[number of live cells.cm-2 of the control] × 100) [63]”

Comment 14

  1. Figure 3: The reason for an additional width for the LDPE should be mentioned in the Figure caption.

Response 14:

Since LDPE had an irregular shape, longest width or length and smallest width were determined.

Comment 15

  1. Figure 11 scale bar missing

Response 15:

 Scale bars have been added to 2 all 2D images. It is not possible to include the exact scale bars in the 3D images.

Comment 16

  1. “. (reviewed in Alijagic et. al.)[78].”

Response 16:

Line 926 has been corrected as,

“MPs are suggested to activate innate immune system upon interaction with cell membrane and/or cellular internalization, potentially leading to activation of inflammasome, a type of immune response mounted following PAMP signaling (reviewed in Alijagic et. al.) [90].”

Comment 17

  1. The authors should also mention, that many novel drug delivery systems will utilize negatively charged polymer systems2and that the particles they investigate might be also get into the body by drinking, or by injecting (particles are also within medical devices like needles and breathing systems). Exposure from these systems also can get into contact with lung cells.

 Response 17:

A sentence has been added on line 72.

“Humans are exposed to inhalable size MPs from not only inhaling aerosolized MPs in the environment but also from polymer based pulmonary drug delivery [25].”

Reference

  1. Rytting E, Nguyen J, Wang X, Kissel T. Biodegradable polymeric nanocarriers for pulmonary drug delivery. Expert Opinion on Drug Delivery. 2008;5(6):629-39; doi: 10.1517/17425247.5.6.629.

Additional changes:

Line 19 has been revised as,

“The toxicity of 11 different types of MPs both commercially purchased, and in-lab prepared MPs was investigated in lung epithelial cells using cell viability, immune and inflammatory response, and genotoxicity endpoints.”

Line 24 has been revised as,

“Of the 11 MPs tested, the small sized Polyethylene Terephthalate (PETE) MPs prepared from disposable water bottles induced the maximum toxicity.”

Line 65 has been revised as,

“Comparatively, MP concentrations in indoor space are high.”

Line 240 has been revised as,

“Water suspended (2-3 µL) MPs were deposited on Calcium fluoride (CaF2) slides (Crystran, Poole, Dorset, UK) and visualized at 10x, 50x and 100x magnification using brightfield illumination.”

Line 350 has been revised as,

“Plates were washed twice and 50 μL of pre-diluted standards or 100 μL of the cell culture supernatants was added to each of the designated wells.”

Line 355 has been revised as,

“The concentration of each chemokine/cytokine was obtained using the Bioplex Manager Software (version 6).”

Moreover, we have realized that Tupperware is a tradename of a plastic produce. Hence the term “Tupperware (PP-TW) has been replaced by polypropylene plastic food storage containers (PP) throughout the manuscript and Figure 1-2, 4-6, 8 as well as the Supplementary Data Files 1-2, Supplementary Information have been revised.

As we added a new Figure S3, the previous Figure S3 – S7 are now Figure S4 – S8.
